# RECURSIVE INTROSPECTION: Teaching Language Model Agents *How* to *Self-Improve*

**Yuxiao Qu**[1]**, Tianjun Zhang**[2]**, Naman Garg**[3]**, Aviral Kumar**[1]
[1]Carnegie Mellon University, [2]UC Berkeley, [3]MultiOn
Website: https://cohenqu.github.io/rise.github.io/, Email: yuxiaoq@andrew.cmu.edu

## Abstract

A central piece in enabling intelligent agentic behavior in foundation models is to make them capable of introspecting upon their behavior, reasoning, and correcting their mistakes as more computation or interaction is available. Even the strongest proprietary large language models (LLMs) do not quite exhibit the ability of continually improving their responses sequentially. In this paper, we develop **RISE:** *Recursive IntroSpEction*, an approach for fine-tuning LLMs to introduce this capability, despite prior work hypothesizing that this capability may not be possible to attain. Our approach prescribes an iterative fine-tuning procedure, which attempts to teach the model how to alter its response after having executed previously unsuccessful attempts to solve a hard test-time problem, with optionally additional environment feedback. RISE poses fine-tuning for a single-turn prompt as solving a multi-turn Markov decision process (MDP), where the initial state is the prompt. Inspired by principles in online imitation and offline reinforcement learning, we propose strategies for multi-turn data collection and training so as to imbue an LLM with the capability to recursively detect and correct its previous mistakes in subsequent iterations. Our experiments show that RISE enables Llama2, Llama3, and Mistral models to improve themselves with more turns on reasoning tasks, outperforming several single-turn strategies given an equal amount of inference-time computation. We also find that RISE scales well, often attaining larger benefits with more capable models, without disrupting one-turn abilities as a result of expressing more complex distributions.

## 1 Introduction

A promising approach to utilizing foundation models, and in particular, large language models (LLMs), is to treat them as general-purpose decision-making machines, or "agents". To be successful, an LLM agent must not just provide plausible completions for input text, but must exhibit interactive, goal-directed behavior to accomplish a given task. Put in abstract terms, this requires mastering two qualities: **(a)** producing responses that explicitly seek information about the task, followed by **(b)** making decisions and improving them by "thinking" and verifying them at inference time. For instance, to succeed in using a new coding library, an effective LLM agent should first synthesize programs, then try the most promising subset against a compiler, use the resulting feedback to improve the program, and repeat the process for multiple turns. Having the ability to successfully improve a response in sequential attempts is equivalent to a form of "self-improvement", at test time.

To enable test-time self-improvement, recent approaches attempt to repurpose the knowledge stored in pre-trained models via few-shot prompting [8, 17, 33, 54, 69]. Although prompt tuning in conjunction with feedback is effective in eliciting improved responses from capable models, it fails to produce models that can succeed in complex tasks by correcting their own mistakes, such as those that require logical reasoning [23, 59]. In many of these problems, models contain the "knowledge" needed to answer a challenging prompt, but fail to elicit that knowledge even when asked to sequentially correct their mistakes. Fine-tuning the LLM on domain-specific question-answering data [7, 31, 41] can help,

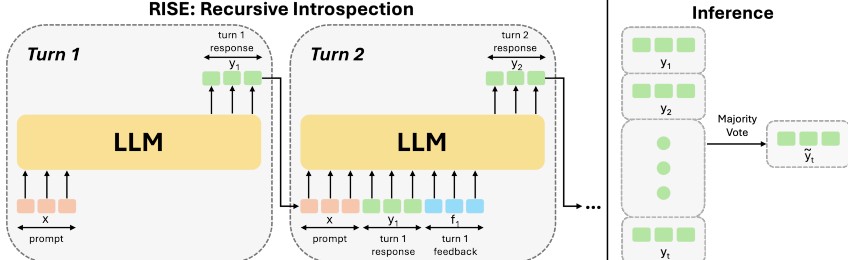

Figure 1: *Recursive Introspection (RISE).* Using multi-round training on on-policy rollouts and supervision from a reward function, RISE trains models that are capable of improving themselves over multiple turns. At inference, we run majority voting on candidate outputs from different turns to obtain the final response.

but it still does not teach the agent a test-time improvement strategy (see Section 5). A strategy for improving responses over sequential attempts at test time is crucial for tackling challenging prompts, where directly attempting the problem in one shot may largely be futile.

**Can we train models to be capable of improving their *own* responses?** If done correctly on a diverse set of problems and scenarios, this could introduce in an LLM, a general procedure for *"how"* it can tackle a hard prompt by improving itself as opposed to supervising it with "what" to respond with, which may not generalize as the test prompt becomes out of distribution. Although one approach to inducing this capability into a model would be to generate data that showcase improvements over multiple sequential turns (potentially from highly capable models), we find that simply imitating these data is not sufficient to enable this capability (Section 5.3). This is due to two reasons: First, multi-turn data from a different model would not show improvements in the kinds of errors the learner would make, thereby being irrelevant to the learner [26]. Second, often sequential multi-turn data collected from proprietary models is also not of high quality since these models are typically not good at proposing meaningful improvements to their own errors [23] even though they can still provide useful responses to the problem at hand. Therefore, we need a different strategy to endow models with a self-improvement capability. Our key insight is to supervise improvements to the learner's own responses in an iterative fashion, taking inspiration from methods in online imitation learning [38] and reinforcement learning (RL) [47]. This supervision can be in the form of oracle responses to the prompt sampled i.i.d. from more capable models, or be generated from the learner itself.

Our contribution, **RISE: Recursive Introspection** (Figure 1), utilizes these insights to improve the self-improvement capability of an LLM over multiple attempts at a given prompt. In each iteration, our approach bootstraps on-policy rollouts from the learner with better responses at the next turn obtained by running best-of-N (using a success indicator on the task) on multiple revision candidates obtained by sampling from the learner itself or using responses from a more capable model, whichever is more convenient. In this way, we construct rollouts that demonstrate the learner how it can improve its responses under its own distribution. Then, we fine-tune the learner on these data using a reward-weighted regression (RWR [36, 37]) objective, that learns from both high- and low-quality parts of such rollouts. By iteratively repeating this procedure, we are able to instill a general self-improvement capability into an LLM. Our results show that LLMs trained via RISE can produce correct responses on more prompts, improving over turns for more challenging prompts.

Even though strong base and intruction-tuned LLMs [25, 63] often fail to improve their responses over multiple sequential attempts (even when explicitly told about their mistakes previously), **RISE** successfully endows similarly-sized LLMs with self-improvement capabilities, resulting in mono-tonically increasing task performance after each turn. Specifically, on the GSM8K [12] dataset, RISE demonstrates significant improvement over various models. RISE improves the performance of LLaMa3-8B by 8.2% and Mistral-7B by 6.6%, entirely using their own data. RISE attains a 17.7% improvement for LLaMa2-7B over the course of 5-turn introspection (outperforming parallel sampling from the first turn), and a 23.9% improvement for Mistral-7B. In contrast, GPT-3.5 itself only improves by 4.6% over five turns. We see similar trends on the MATH dataset [20], where RISE improves LLaMa2-7B by 4.6% and Mistral-7B by 11.1% over five turns. These results consistently demonstrate RISE's effectiveness in enhancing reasoning capabilities for different models.

## 2   Problem Setup and Preliminaries

The goal of our work is to improve LLM performance over sequential attempts / turns at a given problem. Concretely, given a dataset $\mathcal{D} = \{(\boldsymbol{x}_i, \boldsymbol{y}_i^*)\}_{i=1}^{N}$ of problems $\boldsymbol{x}_i$ and oracle responses $\boldsymbol{y}_i^*$,

our goal is to obtain an LLM $\pi_\theta(\cdot|[\boldsymbol{x}, \hat{\boldsymbol{y}}_{1:t}, p_{1:t}])$ that, given the problem $\boldsymbol{x}$, previous model attempts $\hat{\boldsymbol{y}}_{1:t}$ at the problem, and auxiliary instructions $p_{1:t}$ (e.g., instruction to find a mistake and improve the response; or additional compiler feedback from the environment) solves a given problem as correctly as possible. To this end, we encode this goal into the following learning objective:

$$\max_{\pi_\theta} \sum_{i=1}^{L} \mathbb{E}_{\boldsymbol{x}, \boldsymbol{y}^* \sim \mathcal{D}, \hat{\boldsymbol{y}}_i \sim \pi_\theta(\cdot|[\boldsymbol{x}, \hat{\boldsymbol{y}}_{1:i-1}, p_{1:i-1}])} \left[ \mathbb{I}\left( \hat{\boldsymbol{y}}_i == \boldsymbol{y}^* \right) \right]. \tag{2.1}$$

Unlike standard supervised fine-tuning that trains the model $\pi$ to produce a single response $\hat{\boldsymbol{y}}$ given $\boldsymbol{x}$, Equation 2.1 trains $\pi$ to also appropriately react to a given history of responses from its own previous attempts $\hat{\boldsymbol{y}}_{1:i-1}$. Equation 2.1 most closely resembles an RL objective, and we will indeed develop our approach by converting a single-turn problem into a multi-turn MDP. Finally, note that prompting-based methods such as Self-Refine [33] can still be viewed as training $\pi$ to optimize $\pi(\boldsymbol{y}^*|\boldsymbol{x})$ but only when only allowed to modulate the prompt $p_i$ to optimize Equation 2.1. Naturally, since the parameters $\theta$ are unchanged, this would not be effective in optimizing the objective fully.

## 3 RISE: Recursive Introspection for Self-Improvement

Since even strong off-the-shelf models do not exhibit an effective ability to improve themselves when provided with sequential attempts at a given problem [23], a natural next step is to ask how to train models to induce this capability. In this section, we will develop our approach, **RISE**, for fine-tuning foundation models towards improving their own predictions over multiple turns. Our approach will first convert a problem into a multi-turn MDP, then collect data, and finally run offline reward-weighted supervised learning in this multi-turn MDP to induce this capability.

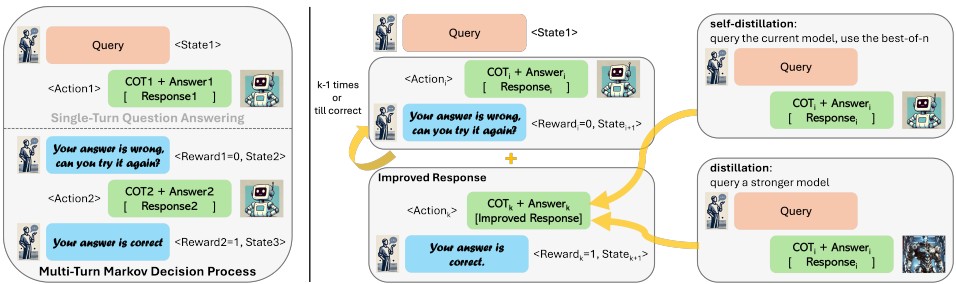

Figure 2: *Left: Problem formulation.* We convert single-turn problems into multi-turn MDPs as discussed in Section 3.1. The state is given by the prompt, history of prior attempts, and optional feedback from the environment. An action is a response generated from the LLM given the state of multi-turn interaction so far. *Right: Data collection.* We collect data by unrolling the current model $k-1$ times followed by an improved version of the response, which is obtained by either (1) **self-distillation**: sample multiple responses from the current model, and use the best response, or (2) **distillation**: obtain oracle responses by querying a more capable model. In either case, RISE then trains on the generated data.

### 3.1 Converting Single-Turn Problems into a Multi-Turn Markov Decision Process (MDP)

The first step in building our approach is to procedurally construct a multi-turn MDP out of a single-turn dataset of prompts and oracle responses (Figure 2, Left). Given a dataset, $\mathcal{D} = \{(\boldsymbol{x}_i, \boldsymbol{y}_i^*)\}$, consisting of prompts $\boldsymbol{x}_i$ and corresponding oracle responses $\boldsymbol{y}_i^*$ (e.g., math questions and natural language responses to those questions), we will construct an *induced* MDP $\mathcal{M}$ from $\mathcal{D}$, and then learn policies in this MDP. An initial state in this MDP is a possible prompt $\boldsymbol{x}_i \in \mathcal{D}$. We denote the output response from the foundation model as action $\boldsymbol{a}$. Given a state $\boldsymbol{s}$, the next state can be obtained by concatenating the tokens representing $\boldsymbol{s}$ with the action $\boldsymbol{a}$ proposed by the model, and an additional fixed prompt $\boldsymbol{f}$ that asks the model to introspect, e.g., *"this response is not correct, please introspect and correct your answer."* (the exact prompt is shown in Appendix F.4). The reward function is a sparse binary indicator of answer correctness at a given state $\boldsymbol{s}$, $r([\boldsymbol{x}_i, \cdots], \boldsymbol{a}) = 1$ if and only if $\boldsymbol{a} = \boldsymbol{y}_i^*$ and is obtained from an answer checking function. This construction from dataset $\mathcal{D}$ to MDP $\mathcal{M}$ is shown below:

$$\mathcal{D} = \{(\boldsymbol{x}_i, \boldsymbol{y}_i^*)\} \rightarrow \mathcal{M}: \rho(\boldsymbol{s}_0) = \text{Unif}(\boldsymbol{x}_1, \boldsymbol{x}_2, \cdots, \boldsymbol{x}_N) \tag{3.1}$$

$$P(\boldsymbol{s}'|\boldsymbol{s}, \boldsymbol{a}) = \delta\left(\boldsymbol{s}' = \text{concat}[\boldsymbol{s}, \boldsymbol{a}, \boldsymbol{f}]\right) \tag{3.2}$$

$$r(\boldsymbol{s}, \boldsymbol{a}) = \mathbf{1}\left(\boldsymbol{a} = \boldsymbol{y}_i^* \text{ if } \boldsymbol{x}_i \in \boldsymbol{s}\right). \tag{3.3}$$

## 3.2 Learning in the Multi-Turn MDP

With the MDP construction in place, the next step involves training a model to improve itself over the course of a rollout. We subscribe to an offline approach to learning that we describe in the following.

**Step 1: Data collection for self-improvement.** To ensure that rollout data from this multi-turn MDP is useful for teaching the model how to self-improve, it must satisfy a few desiderata: **(1)** it must illustrate the mistakes that the learner is likely to make and showcase how to improve upon them in the next attempt, **(2)** the data must illustrate responses that are relevant to the model given the problem and previous attempts in context, and **(3)** it must not contain any rollout that degrades in a subsequent turn. Our data collection strategy (Figure 2, Right) satisfies these desiderata.

In a given round $k$, for a given problem $\boldsymbol{x}_i$, we unroll the *current* model $\pi_{\theta_k}(\cdot|\cdot)$ to produce multiple sequential attempts, denoted by $\boldsymbol{y}_t^i \sim \pi_{\theta_k}(\cdot|\boldsymbol{s}_t^i)$. In problems, where external input (e.g., compiler feedback) is available, we also observe a variable-length, natural language external input, $f_t^i$ (e.g., in math problems we ask the model to correct itself). We also observe a scalar reward value $r(\boldsymbol{s}_t^i, \boldsymbol{y}_t^i)$, denoted as $r_t^i$ in short. Let us denote this dataset of "on-policy" model rollouts as $\mathcal{D}_{\text{on-policy}} := \{(\boldsymbol{s}_t^i, \boldsymbol{y}_t^i, f_t^i, r_t^i)_{t=1}^T\}$.

For each time-step, we construct an improved version of the response $\boldsymbol{y}_t^i$ that we will denote by $\tilde{\boldsymbol{y}}_t^i$. We also record the reward score associated with this improved response as $r(\boldsymbol{s}_t^i, \tilde{\boldsymbol{y}}_t^i)$, or $\tilde{r}_t^i$ in short. To obtain an improved version of a response $\boldsymbol{y}_t^i$, we can employ several strategies. Perhaps the most straightforward approach is to query an off-the-shelf more capable model to provide a correct response given the prompt $\boldsymbol{x}_i$, the previous response $\boldsymbol{y}_t^i$, and an optional external feedback $f_t^i$. We refer to this as the **distillation** variant of our approach, since it uses a strong "teacher" model to guide self-improvement (note that this is different from the classic notion of knowledge distillation, and we will in fact show results in Section 5.1 that will help understand the differences).

$$\tilde{\mathcal{D}}_{\text{on-policy + distill}} := \left\{ \left\{ \left( \boldsymbol{s}_t^i, \tilde{\boldsymbol{y}}_t^i, f_t^i, \tilde{r}_t^i \right) \right\}_{t=1}^T \right\}_{i=1}^{|\mathcal{D}|}. \tag{3.4}$$

The second variant of our approach, which alleviates the need for a teacher model, involves constructing an improved response by sampling multiple times from the learner itself. We refer to this approach as the **self-distillation** variant. Concretely, for each state in the dataset, $\boldsymbol{s}_t^i \in \mathcal{D}_{\text{on-policy}}$, we sample $N$ responses $\tilde{\boldsymbol{y}}_t^i[0], \tilde{\boldsymbol{y}}_t^i[1], \cdots, \tilde{\boldsymbol{y}}_t^i[N] \sim \pi_\theta(\cdot|\boldsymbol{s}_t^i)$, and use the best response from these $N$ candidates (as measured by the associated reward values $\tilde{r}_t^i[0], \cdots, \tilde{r}_t^i[N]$) to relabel the model response at the next step $t+1$ in an improvement trajectory. Formally, say $\tilde{\boldsymbol{y}}_t^i[m] = \arg\max_{j \in [N]} r(\boldsymbol{s}_i, \tilde{\boldsymbol{y}}_t^i[j])$, then we label the responses in the dataset $\mathcal{D}_{\text{on-policy}}$ at step $t+1$ with the improved response and its associated reward value $\tilde{r}_t^i[m]$:

$$\tilde{\mathcal{D}}_{\text{on-policy + self-distillation}} := \left\{ \left\{ \left( \boldsymbol{s}_{t+1}^i, \tilde{\boldsymbol{y}}_t^i[m], f_{t+1}^i, \tilde{r}_t^i[m] \right) \right\}_{t=0}^{T-1} \right\}_{i=1}^{|\mathcal{D}|}. \tag{3.5}$$

**Step 2: Policy improvement.** With the aforementioned data construction schemes, we can now train a model on these datasets. While in general, any offline RL approach can be used to train on these data, in our experiments we adopt an approach based on weighted supervised learning [37] due to ease of experimentation and its simplicity. In particular, we perform a weighted supervised regression, where the weights are given by the exponential transformation of the reward values in $\tilde{\mathcal{D}}$.

$$\text{Reward-weighted RL:} \quad \max_\theta \ \mathbb{E}_{\boldsymbol{x}_i \sim \tilde{\mathcal{D}}} \left[ \sum_{t=1}^T \log \pi_\theta(\tilde{\boldsymbol{y}}_t^i|\boldsymbol{s}_t^i) \cdot \exp(r_i^t/\tau) \right], \tag{3.6}$$

where $\tau$ is a temperature parameter to further expand or narrow the difference between good and bad actions. In our preliminary experiments, we found that Equation 3.6 can often induce a bias towards increasing log likelihoods of responses where rewards are high, prioritizing updates on easy problems where rewards are already high. To address this issue, we apply a slight modification to Equation 3.6 and center the exponentiated rewards around the mean value averaged across all attempts on a given prompt, akin to advantage-weighted regression [36]. We find that the use of advantages in place of rewards helps us avoid the "rich-gets-richer" phenomenon with easy problems.

## 3.3 Inference at Deployment Time

RISE can be run in two modes at inference time. The most straightforward way to run the policy $\pi_\theta(\cdot|\cdot)$ trained by RISE is within a multi-turn rollout, where the model samples a new response

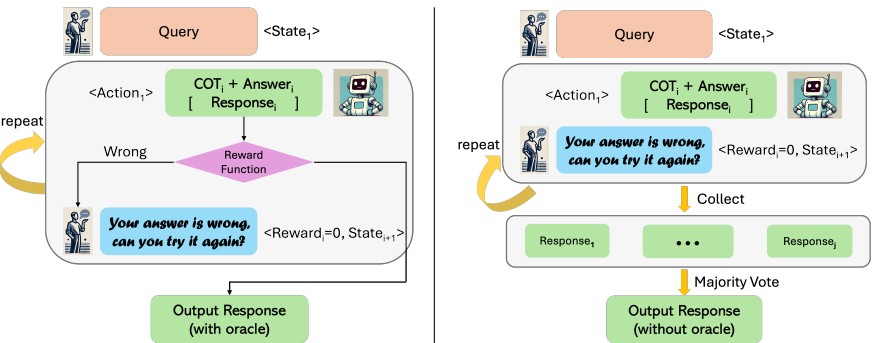

Figure 3: *RISE Inference.* There are two ways to query the model trained via RISE upon inference: **(1)** **with oracle** (*Left*): each time the model improves its response, it is allowed to check its answer against an environment and terminate early as soon as a correct answer is found; or **(2) without oracle** (*Right*): we ask the model to sequentially revise its own responses j times, and perform majority voting on all candidate outputs from different turns to obtain the final response. If the turn number $j$ is larger than the iteration number $k$, the agent only keeps the most recent history with $k$ interactions to avoid test-time distribution shift.

conditioned on the past context (i.e., state in the multi-turn MDP). This past context consists of the external feedback $p_i^{\text{test}}$ concerning the response $\boldsymbol{y}_i^{\text{test}}$ and the rollout terminates as soon as the current response is judged to be correct according to the environment's answer verification function. In other words, we terminate the rollout as soon as the reward equals to the reward for the oracle response: $r(\boldsymbol{x}, \boldsymbol{y}_i^{\text{test}}) = r(\boldsymbol{x}, \boldsymbol{y}^*)$. This protocol queries the reward function after each turn in the rollout. Since several reward function queries are performed, we refer to this approach as **"with oracle"**.

RISE can also be run in a mode that avoids the need to query the answer checker or the reward function within a rollout. In this case, we run full-length rollouts by forcing the model to retry, ignoring the correctness of the response. We then utilize a self-consistency mechanism [53] based on majority voting to decide the candidate response at the end of each turn. Concretely, at the end of each turn $j$, we identify the response by running a majority vote over all response candidates from the previous turns $(\text{maj}\left(\boldsymbol{y}_1^{\text{test}}, \boldsymbol{y}_2^{\text{test}}, \cdots, \boldsymbol{y}_j^{\text{test}}\right))$, including turn $j$. We call this **"without oracle"**. A schematic illustration of these approach is shown in Figure 3. Most of our evaluations use no oracle.

At iteration $k$, since the agent is able to improve its own response from $j$ to $j + 1$ when $j \leq k$, to avoid test time distribution shift, in both modes, we use a size $k$ shift window to store the most recent conversation history when the turn number $j$ is larger than the iteration number $k$.

### 3.4 Practical Algorithm and Implementation Details

A complete algorithmic pseudocode for each approach is shown in Appendix D. We trained 7B models via RISE and found that these models often could not adhere to response style and instructions for improving their responses when generating on-policy data. As a result, before running on-policy data collection, we find it often useful to run an initial phase of supervised fine-tuning on in-domain, multi-turn rollouts generated from a capable model to provide style and instruction-following information to the learner. We call this the "knowledge boosting" stage. We then run on-policy rollouts starting from a boosted model. In each iteration, we generate 1 trajectory for each unique problem. We then run fine-tuning, with hyperparameters and details in Appendix F. For iterative fine-tuning, we find that starting from the *base* model but training on data from all iterations thus far is more beneficial than continued fine-tuning from the checkpoint obtained in the previous iteration.

## 4 When and Why is Self-Improvement Over Turns Possible?

A natural question to ask is why self-improvement with **RISE** even possible. One might surmise that the model may simply not have enough knowledge to correct its *own* mistakes if it is unable to correctly answer the problem in the first turn. Then, why is it possible to teach the model to correct its own mistakes? In this section, we provide the reason why this kind of self-improvement is possible, supported with empirical evidence to justify our hypotheses.

Iteratively teaching a model how to make updates on a given response can be crucial when representing the target distribution $p^*(\boldsymbol{y}|\boldsymbol{x})$ requires more capacity than what the model $\pi_\theta$ affords by conditioning on only the input prompt tokens. When the target distribution requires greater capacity, learning a sequence of conditionals, $\pi_\theta(\boldsymbol{y}_{i+1}|\boldsymbol{x}, \boldsymbol{y}_{0:i})$ followed by marginalization is expected to induce a more flexible marginal distribution over $\boldsymbol{y}_T$ given $\boldsymbol{x}$. This hypothesis is akin to the difference between

diffusion models [44] and variational autoencoders (VAEs) [27] in image generation: iteratively fitting a sequence of generative distributions over intermediate noisy inputs in a diffusion model gives rise to a more flexible distribution [45] than monolithic variational auto-encoding, even though diffusion models still utilize an evidence lower-bound objective(ELBO). While the diffusion process utilizes hand-designed noise schedules, RISE utilizes the base model itself to induce iterative improvements.

To verify this hypothesis, we tracked the training un-weighted, negative log-likelihood loss (NLL) values for the oracle response $y^*$ given the input prompt $x$ marginalized over intermediate steps in a multi-turn rollout, and compared it against the NLL values $-\log p_\theta(y^*|x)$ attained by directly attempting to predict the final response in Figure 4 (Left, labeled as "Classic"). Concretely, we sampled 256 prompts $x$ and their oracle responses $y^*$ and computed the average $-\log p_\theta(y^*|x)$ across all $x$, along with a 95% confidence interval for

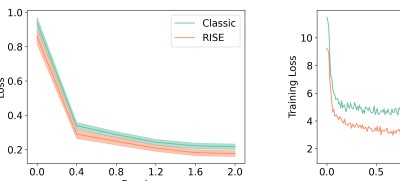

Figure 4: *Left: The probability of the true answer given the prompt.* Observe that model trained with RISE has higher probability for the true answer. **Right: The training perplexity (loss) of fitting only the oracle answer or a sequence of answers.** Note that fitting a sequence of answers (RISE) reduces the loss more than fitting the oracle answer (Classic).

different checkpoints during training. We find that for any number of epochs (including fractional number of epochs on the x-axis), the NLL value is lower when conditioning on multi-turn data that RISE generates compared to oracle responses to the prompts obtained from an expert. This suggests that RISE is able to utilize the computation of tokens from previous turns to model the target distribution. We also measure the average NLL loss on all samples through training, sampled i.i.d. from the training dataset for RISE and classic fine-tuning and observe a similar trend: RISE reduces loss more than the standard approach, attaining lower perplexity values (Figure 4 Right).

For knowledge-based question answering, it is not possible for the model to produce any meaningful improvements because learning $p^*(y|x)$ is not bounded by insufficient capacity of $\pi_\theta(y|x)$, but is rather unable to match $p^*$ due to the absence of features that are critical to learn the correct mapping from $x$ to $y$. Training with RISE would likely incentivize hallucinations [26], since more input tokens appearing from previous attempts would only provide easier ways to pick up on spurious correlations. However, this is not the failure mode on reasoning problems [29], where maj@K rates at turn 1 tend to be higher than pass@1 as we find in our experiments (indicating that performance can be improved by sampling the model itself). Figure 5 demonstrates that RISE's sequential procedure solves many problems unsolved by pass@B for larger $B$ in the first turn, indicating it learns to access the model's pre-trained knowledge differently, rather than merely converting pass@K to pass@1 performance like most single-turn approaches.

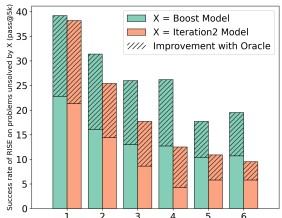

Figure 5: *Fraction of problems unsolved by pass@B at first turn that sequential 5-turn RISE sampling solves,* where $B = 5 \times k$ ($k$ is the x-axis). RISE can solve several challenging problems that sampling at the first turn with much larger budgets cannot solve.

## 5 Experimental Evaluation

The goal of our experiments is to demonstrate the efficacy of RISE in instilling language models with the ability to self-improve their responses over turns. Our experiments answer the following questions: **(1)** How effectively can RISE improve performance over multiple sequential attempts (i.e., turns) at a given prompt?; **(2)** Does the performance of RISE improve with more rounds of iterative training?; **(3)** Does the self-improvement strategy induced by RISE generalize to novel problems that are out of the training domain? and finally; **(4)** What is the best data composition for training RISE? To this end, we compare RISE to other prior and baseline approaches, and perform ablations on GSM8K [12], MATH [20]. We defer RISE's performance on code generation tasks to Appendix B.3. Additional analyses are presented in Appendix C, where we examine the out-of-distribution generalization capabilities of RISE and explore the model's behavior in correct-to-correct scenarios. We also present a detailed study of weak-to-strong generalization, demonstrating how RISE-generated data from weaker models can enhance the performance of stronger models.

**Baselines, comparisons, and evaluation.** We compare RISE to several prior methods that attempt to induce similar self-improvement capabilities: **(a) self-refine** [23, 33] that prompts a base model to critique and revise its mistakes; **(b) GloRE** [19], which trains a separate reward model to locate

errors and a refinement model to improve responses of a base LLM; and **(c) self-consistency** [53], which runs majority voting on multiple responses from the first turn as a baseline to compare to our sequential strategy. We tried to construct fair comparisons between RISE and these methods using a similar-sized model [25, 63], but differences in base model, training data, and evaluation setups still prohibits us from performing an apples-to-apples comparison in some cases. Nonetheless, we can still understand the ballpark of improvement by contextualizing our results with these prior works. We also compare to V-STaR [21], but since this is not an fair comparison, we defer it to Appendix B.1.

We evaluate RISE in both modes at inference time: with and without an oracle (Section 3.3) at the end of five turns (the performance for each turn is in Appendix B.2). Concretely, these metrics are defined as follows: **(1) with oracle, "p1@t5"**: this run terminates the rollout as soon as the response is correct. In other words, this metric allows queries to the final answer verifier at the end of each turn, and **(2) without oracle, "m1@t5"**: this run sequentially sample five responses, and we compute the maj@1 performance on the candidates produced in each turn as detailed in Section 3.3. We also compare maj@K performance at the first turn for all the models we train.

| Approach | GSM8K [11] w/o oracle | | | w/ oracle | MATH [20] w/o oracle | | | w/ oracle |
|---|---|---|---|---|---|---|---|---|
| | m1@t1 | → m5@t1 | → m1@t5 | p1@t5 | m1@t1 | → m5@t1 | → m1@t5 | p1@t5 |
| **RISE (Ours)** | | | | | | | | |
| Llama2 Base | 10.5 | 22.8 (+12.3) | 11.1 (+0.6) | 13.9 (+3.4) | 1.9 | 5.1 (+3.2) | 1.4 (-0.5) | 2.3 (+0.4) |
| +Boost | 32.9 | 45.4 (+12.5) | 39.2 (+6.3) | 55.5 (+22.6) | 5.5 | 6.8 (+1.3) | 5.5 (0.0) | 14.6 (+9.1) |
| +Iteration 1 | 35.6 | 49.7 (+14.1) | 50.7 (+15.1) | 63.9 (+28.3) | 6.3 | 8.8 (+2.5) | 9.7 (+3.4) | 19.4 (+13.1) |
| +Iteration 2 | 37.3 | 51.0 (+13.7) | 55.0 (+17.7) | 68.4 (+31.1) | 5.8 | 10.4 (+4.6) | 10.4 (+4.6) | 19.8 (+14.0) |
| **SFT on oracle data** | | | | | | | | |
| Only correct data | 27.4 | 42.2 (+14.9) | 34.0 (+6.6) | 43.6 (+16.2) | 5.8 | 7.9 (+2.1) | 5.5 (-0.3) | 12.1 (+6.2) |
| Correct and incorrect | 25.7 | 41.8 (+16.1) | 31.2 (+5.5) | 41.5 (+15.8) | 5.0 | 5.2 (+0.2) | 5.0 (+0.0) | 13.1 (+8.1) |
| **RISE (Ours)** | | | | | | | | |
| Mistral-7B | 33.7 | 49.4 (+15.7) | 39.0 (+5.3) | 46.9 (+13.2) | 7.5 | 13.0 (+5.5) | 8.4 (+0.9) | 13.0 (+5.5) |
| + Iteration 1 | 35.3 | 50.6 (+15.3) | 59.2 (+23.9) | 68.6 (+33.3) | 6.7 | 9.5 (+2.8) | 18.4 (+11.1) | 29.7 (+22.4) |
| **7B SoTA [63]** | | | | | | | | |
| Eurus-7B-SFT | 36.3 | 66.3 (+30.0) | 47.9 (+11.6) | 53.1 (+16.8) | 12.3 | 19.8 (+7.5) | 16.3 (+4.0) | 22.9 (+10.6) |
| **Self-Refine [33]** | | → m1@t3 | → p1@t3 | | | → m1@t3 | → p1@t3 |
| Base | 10.5 | 22.4 (+11.9) | 7.1 (-3.4) | 13.0 (+2.5) | 1.9 | 5.1 (+3.2) | 1.9 (0.0) | 3.1 (+1.2) |
| +Iteration 2 | 37.3 | 50.5 (+13.2) | 33.3 (-4.0) | 44.5 (+7.2) | 5.8 | 9.4 (+3.6) | 5.7 (-0.1) | 9.5 (+3.7) |
| GPT-3.5 | 66.4 | 80.2 (+13.8) | 61.0 (-5.4) | 71.6 (+5.2) | 39.7 | 46.5 (+6.8) | 36.5 (-3.2) | 46.7 (+7.0) |
| Mistral-7B | 33.7 | 48.5 (+14.8) | 21.2 (-12.5) | 37.9 (+4.2) | 7.5 | 12.3 (+4.8) | 7.1 (-0.4) | 11.4 (+3.9) |
| Eurus-7B-SFT | 36.3 | 65.9 (+29.6) | 26.2 (-10.1) | 42.8 (+6.5) | 12.3 | 19.4 (+7.1) | 9.0 (-3.3) | 15.1 (+2.8) |
| **GloRE [19]** | | → m1@t3 | → p1@t3 | | | | |
| +ORM | 48.2 | | 49.5 (+1.3) | 57.1 (+8.9) | | | | |
| +SORM | 48.2 | | 51.6 (+3.4) | 59.7 (+11.5) | | Not studied in [19] | | |
| +Direct | 48.2 | | 47.4 (-0.8) | 59.2 (+11.0) | | | | |

Table 1: **RISE vs. other approaches (Self-Refine, GLoRE) and baselines.** Observe that RISE attains the biggest performance improvement (in brown) between 1-turn (m5@t1) and 5-turn (m1@t5) performance w/o an oracle on both GSM8K and MATH. This performance gap grows larger when oracle early termination is allowed (p1@t5 w/ oracle). Self-Refine [33] degrades performance across the board when used without an oracle, and attains minor performance improvements with an oracle. GLoRE trains a separate refinement model, but performs worse than RISE; more details about it are in Appendix B.1. Using RISE on top of a better base model (Mistral-7B) is also effective (positive improvements with multiple turns), and the m1@t5 performance of Mistral-7B exceeds even state-of-the-art math models such as Eurus-7B-SFT [63]. Simply running single-turn SFT on data utilized by RISE is not effective at inducing a self-improvement capability, implying that algorithmic design choices in RISE are crucial for performance. Color coding indicates numbers that can be compared.

## 5.1 Does RISE improve performance over multiple turns compared to other approaches?

**Main results.** We present the comparisons in Table 1. First, note that RISE ("Iteration 1" and "Iteration 2") boosts up the LLama2 base model's five-turn performance by 15.1% and 17.7% respectively with each iteration on GSM8K and 3.4% and 4.6% on MATH, w/o any oracle. Interestingly, we found using prompting-only self-refine [33] largely degrades performance across the board, even with a strong proprietary model, GPT-3.5. The strongest 7B base models, Mistral-7B and Eurus-7B-SFT [63], when coupled with standard prompting, are only able to improve their performance, but only by 5.3% / 11.6% and 0.9% / 4.0% respectively on GSM8K and MATH, which is significantly lower than our approach. The performance of GLoRE improves only by 3.4% on GSM8K (over two turns), but this is still lower than our approach, which improves by 6.3% in two turns and 13.4% in three turns (see Appendix B.1). This indicates that RISE is effective in teaching models how

to improve their own errors. **To summarize,** training with RISE gives the largest performance improvement gains compared to other approaches both with and without the use of an oracle, and these gains are transferred to other base models.

One might also hypothesize that the performance gains with RISE here are largely a result of utilizing queries to an off-the-shelf more capable model for providing supervision and not the algorithmic approach for data collection and training. To address this hypothesis, we store all the data generated by RISE from more capable models and train on this data via standard single-turn SFT ("**SFT on oracle data**"). Since not all of this data are guaranteed to be correct, we also run this experiment on only the correct responses in these oracle data. Observe in Table 1 that this procedure does not still instill self-improvement capabilities, largely preserving or degrading sequential ("**maj@1@turn5**") performance compared to simply sampling one response in the first turn. This means that the algorithmic design of RISE is critical, as opposed to the use of expert supervision.

### 5.1.1 Can RISE Effectively Make Use of Mistakes and Correct Them?

One concern that arises from prior results on self-refinement or self-correction is whether the model can truly correct itself over turns or whether the improvement comes from the effect of sampling more answers and picking the best one. In Table 1, we see that sequentially improving responses via RISE ("**maj@1@turn5**") outperforms sampling 5 responses in parallel at the first turn and applying a majority vote on them ("**maj@5@turn1**"). Please note that this comparison utilizes an equal number of samples, with the only difference being that these samples are drawn in parallel at the first turn in one case and sequentially at the end of five turns in the other. Comparing maj@5 performance at the end of 1 turn and 5 turns, we observe a consistent 4% to 8% improvement on GSM8K and an 6.5% improvement on MATH (with Mistral-7B model). This means that RISE can imbue models with a self-improvement ability, while running parallel sampling alone on any model cannot endow the same ability. Even the **maj@5@turn1** performance of standard single-turn SFT on the data used by RISE is substantially worse than the sequential **maj@1@turn5** performance of RISE, implying that the algorithmic protocol of RISE plays a critical underlying role. Finally, we also remark that in Figure 5, we showed that the sequential procedure learned by RISE over five turns could solve a significant fraction of problems that were unsolved by pass@B for much larger values of $B \gg 5$ in the first turn, implying that sequential RISE can actually tackle prompts that were not solvable by simply sampling more responses in the first turn. One might speculate if these improvements in sequential improvement ability largely come at a cost of reduced improvements in first turn performance. We also observe that running multiple iterations of RISE still preserves the first turn performance while improving the 5-turn performance.

### 5.1.2 How Does the Base Model Affect RISE?

The performance of RISE with Llama2-7B on an absolute scale is lower than the best models specifically fine-tuned on math data (e.g., Eurus-7B-SFT or Mistral-7B). However, we find that RISE is still effective on top of Mistral-7B base model. In fact, *our performance at the end of five turns outperforms one of the best 7B SFT models, customized to math reasoning.* Compare the m1@t5 performance of Eurus-7B-SFT and Mistral-7B in RISE (ours), to find that Mistral-7B + RISE outperforms Eurus-7B-SFT.

### 5.1.3 Self-Distillation Version of RISE

We compare the performance of RISE with self-generated data and supervision (Equation 3.4, $N = 16$) after one iteration directly on top of more capable models: Mistral-7B and Llama-3-8B on GSM8K in Table 2, without any knowledge boosting phase. We find that this variant also improves the 5-turn performance of the base model compared to the first turn: compare "m1@t5" vs "m1@t1" for both the models Llama-3-8B and Mistral-7B, where RISE boosts the sequential self-improvement performance by more than 1% compared to turn 1 performance w/o any oracle.

| RISE (Self) | w/o oracle | | | w/ oracle |
|---|---|---|---|---|
| | m1@t1 | → m5@t1 | → m1@t5 | p1@t5 |
| Mistral-7B | 33.7 | 49.4 (+15.7) | 39.0 (+5.3) | 46.9 (+13.2) |
| + Iteration 1 | 36.8 | 44.4 (+7.6) | 39.5 (+6.6) | 48.7 (+15.9) |
| Llama-3-8B | 45.3 | 69.7 (+24.4) | 52.5 (+7.2) | 61.0 (+15.7) |
| + Iteration 1 | 65.6 | 80.7 (+15.1) | 73.8 (+8.2) | 81.2 (+15.6) |

Table 2: ***RISE with self-distillation on GSM8K.*** RISE is able to improve 5-turn maj@1 performance of the model with entirely self-generated data and supervision, despite the fact that the base Mistral-7B model does not produce correct answers for several problems.

Of course, this version of RISE does not outperform the "m5@t1" performance of the fine-tuned model. We expect this to be largely a function of one single iteration of training. Since the self-distillation version of RISE utilizes best-of-N sampling against the same model to produce supervision

for self-improvement, RISE would first have to match the performance of best-of-N sampling before it can start to improve over it via reward maximization. Due to the significant gap between the base model's m5@t1 and m1@t5 performance, we expect that this will take quite a few iterations or a fully online RL algorithm. We did not have computational resources and infrastructure to run multiple iterations, but this is an interesting avenue for future work. In this self-distillation setting, we could also divide the computation between sequential and parallel sampling strategies to get the best results at the end of five turns. Nonetheless, this result shows that even by training on self-generated samples, RISE can actually amplify the sequential sampling performance of the base model.

## 5.2 Does the Performance of RISE Improve with Iterative Training?

Next, we attempt to understand if RISE improves with multiple rounds of training on on-policy data. As shown in Tables 1 and 2, the performance of RISE improves from iteration to iteration constantly. The 5-turn performance of RISE, both with and without an oracle, exhibits a clear improvement with more rounds. This implies that iterative self-training procedures of the form of STaR [66] can also be combined with RISE to train models for self-improvement. This also perhaps serves as a strong hint towards the potential utility of full online reinforcement learning (RL) techniques.

## 5.3 What Data Compositions and Data Quantity are Crucial for RISE?

We now study how different data compositions affect the performance of RISE with the goal of answering questions such as *should we collect on-policy error correction data like DAgger [38] or should we bias towards high-quality off-policy data?*. To understand the utility of different data compositions, we enlist the three aspects RISE: **(a)** the use of multi-turn rollout data for fine-tuning, **(b)** the use of unsuccessful / suboptimal rollouts via weighted supervised fine-tuning compared to naïve supervised learning, which only utilizes successful rollouts for fine-tuning; and **(c)** the use of on-policy rollouts and self-generated or oracle data. We will now perform controlled experiments to understand the effect of each of these factors on the overall performance of RISE.

**(a) Data composition for fine-tuning.** We first study the necessity of using the interaction of error correction history in RISE in Figure 6 (Left). We compare two approaches: model trained with oracle answers shown right after the query ("1-turn") and oracle answers shown after intermediate failed attempts ("Multi-turn") in Figure 6 (Left). Even though the latter trains on intermediate responses that may not always be correct, it attains a higher performance than training on the correct response for a given prompt. This highlights the importance of training on contexts with a multi-turn interaction history depicting mistakes from the learner to improve self-improvement capabilities.

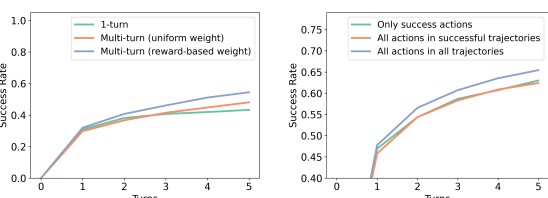

Figure 6: *Left: The importance of multi-turn interaction history and weighted objectives for training RISE.* Note that training with multi-turn data leads to better self-improvement performance at the end of 5 turns, than one-turn data from the original dataset with oracle answers from another model; also observe that using a weighted objective performs better. *Right: The importance of using all rollouts for learning*, instead of only successful rollouts or only successful responses in the data. Using all data performs best in our results.

**(b) Weighted supervised learning vs unweighted supervised learning.** We examine reward-weighted RL's impact on multi-turn data in RISE as opposed to simply imitating filtered successful data. We find that using all the data leads to improved performance over simply filtering good data in Figure 6 (Right), which reduces sample size. In Figure 6 (Left), we find that reward-weighted training improves performance on later turns, allowing us to better leverage all the sub-optimal data.

**(c) On-policy vs off-policy data; self-generated vs. expert data.** RISE runs on-policy rollouts and seeks improvements on responses that the learner produces. As shown in Figure 9 (Left), a "DAgger [38]"-style approach that seeks improvements on responses appearing in on-policy rollouts improves performance (green/orange) compared to using expert data alone (blue/pink). Conceptually, this addresses the train-test mismatch between the distribution of context tokens, enabling imitation learning methods to now target the correct distribution. In addition, recent work [26] has shown that LLMs often memorize "unfamiliar" examples generated by oracle models; by training on on-policy rollouts, we should be able to eliminate any such potential issues. Thus, while the model trained via offline imitation is able to reduce loss, these improvements do not generalize to new problems.

## 5.4 Error Analysis of RISE over Turns

Following the protocol of Huang et al. [23], in this section, we perform an error analysis of the improvement performed by RISE (without any oracle feedback) to understand how the fraction of incorrect and correct responses changes over turns, when **no oracle** is used for early termination. We demonstrate this in the form of Venn diagrams in Figure 7. First note that there is a consistent increase in the portion of problems that stay correct and a consistent decrease in the portion of problems that stay incorrect, which means that the model is able to answer more and more problems as we increase the number of turns. Second, there is a consistent decrease in the number of problems that change from being correct to incorrect, which is often also not the case for strong proprietary LLMs such as GPT in Huang et al. [23]. We also note that there is a decrease in the total number of incorrect problems that become correct in the subsequent turn, but this is a direct consequence of a shrinkage in the size of the incorrect response set as more problems become correct over turns. This indicates that one can induce "intrinsic" self-improvement (per the terminology of Huang et al. [23]) via fine-tuning with RISE, even though no external environment input is provided during evaluation.

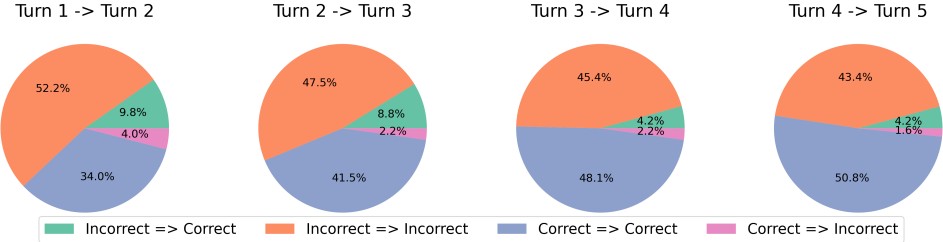

Figure 7: *Change in the fraction of responses that transition their correctness values over the course of multi-turn rollouts from RISE, w/o oracle.* Observe that in general, the fraction of Correct → Correct responses increases; Incorrect → Incorrect responses decreases; and the fraction of Correct → Incorrect responses also decreases, indicating that RISE (w/o any oracle) is able to iteratively improve its responses.

**Qualitative examples.** We also inspect several examples from the GSM8K test set to qualitatively understand the behavior of RISE over turns and observe different behavior patterns, that we show in Appendix E. For instance, the trained model may choose to completely rewrite its previous response if it is totally incorrect in order to get to the correct answer or make small edits if the previous response is mostly correct. Another interesting pattern we note is that the model implicitly has the ability to locate errors in previous responses and only refine the erroneous steps. Additionally, the model is tolerant of noisy environmental feedback when there is no oracle-assisted early termination.

## 6 Discussion, Future Directions, and Limitations

We presented RISE, an approach for fine-tuning LLMs to be able to improve their own responses over multiple turns sequentially. RISE prescribes an iterative RL recipe on top of on-policy rollout data, with expert or self-generated supervision to steer self-improvement. RISE significantly improves the self-improvement abilities of 7B models on reasoning tasks (GSM8K and MATH), attaining an improvement over turns that previous work [23] has not observed in strong proprietary models. In addition, RISE outperforms prior approaches that attempt to tackle similar problems of refinement and correction, while being simpler in that it does not require running multiple models and works well with just one model.

Despite these good results, there are still many open questions and limitations. Due to computational constraints, we were not able to perform more than two iterations of training with RISE, and no more than one iteration when the supervision comes from the learner itself. Improving with self-generated supervision will likely require more computation and more iterations, since it will be slower than when using an off-the-shelf expert model. RISE requires running manual iterations and hence, a more "online" variant of RISE is likely the solution in the long run, especially when we wish to scale on-policy learning in a data-efficient manner. Additionally, while our work fine-tunes models on one task at a time, it will be certainly interesting to include data from the protocols specified by RISE into general instruction tuning and post-training pipelines. Given the results that fine-tuning on data prescribed by RISE does not hurt the first-turn performance of any model we trained, we hypothesize that adding this sort of data in general instruction-tuning pipelines should not hurt either, while enabling the sequential self-improvement capability that is largely absent from models today.

## Acknowledgements

This work was done at Carnegie Mellon University. We thank Fahim Tajwar, Abitha Thankaraj, Amrith Setlur, and Charlie Snell for their feedback and informative discussions. This work was supported by ONR under N000142412206, OpenAI superalignment fast grants, and used the Delta system and JetStream2 [18] at the National Center for Supercomputing Applications through CIS240249 and CIS230278, supported by the National Science Foundation. We thank OpenAI for providing GPT-4 credits for academic use.

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

# Appendices

## A    Related Work

Several prior works build techniques to improve reasoning and thinking capabilities of foundation models for downstream applications. Typically these works focus on building prompting techniques for effective multi-turn interaction with external tools [6, 8, 16, 34, 51, 58, 60], sequentially refining predictions by reflecting on actions [8, 17, 68], asking the model to verbalize its thoughts [35, 54, 70], asking the model to critique and revise itself [33, 42] or by using other models to critique a primary model's responses [3, 14, 22, 58]. Although a subset of this work does improve its own responses, this self-correction ability often requires access to detailed error traces (e.g., execution traces from code compilers [8, 33]) in order to succeed. In fact, [23] and Table 1 both indicate that self-improvement guided by the LLM itself (i.e., "intrinsic self-correction") is often infeasible for off-the-shelf LLMs even when they contain the knowledge required to tackle the prompt given, **but fine-tuning with RISE induces this capability** as we show in this paper.

Beyond prompting, previous work also attempts to fine-tune LLM to obtain self-improvement capabilities [7, 41, 67]. These works attempt to improve reasoning performance by training on self-generated responses [32, 48, 62, 63, 65]. To achieve this, these works use a combination of learned verifiers [13, 30, 49, 52], search [15, 28, 35, 40], contrastive prompting on negative data [10, 50], and iterated supervised or reinforcement learning (RL) [9, 39, 57, 64]. Although our approach also trains on model-generated data, we aim to introduce a complementary capability to improve performance over sequential turns of interaction, rather than to improve single-turn performance alone. Other work fine-tunes LLMs for multi-turn interaction directly via RL [43, 71]: while this is indeed related, single-turn problems posed in multi-turn scenarios require addressing distinct challenges than generic multi-turn RL: **(i)** sample-efficiency is not a concern since the entire environment is fully characterized by the training dataset of prompts and oracle answers and dynamics are deterministic, and **(ii)** we need to generalize to novel test prompts. Multi-turn RL focuses on sample efficiency, which is not as critical in our setting, though of course learning to generalize from a limited number of initial states would be appealing. Our main focus is to show that it is possible to train models for self-improvement via appropriately designing multi-turn fine-tuning objectives. This is orthogonal from the choice of training approach (RL or not).

The most related to our work are GLoRE [19] and Self-Correct [55], which train separate models to identify errors and refine incorrect answers of other LLMs. Unlike these works, our approach trains a single model to produce answers and improve them over more than two turns, which is the maximal number of turns studied in these works. We show that doing so successfully requires careful design choices: an iterative on-policy data generation strategy along with a training objective that can learn from both successful and unsuccessful rollouts. From an algorithmic point of view, RISE is similar to online imitation learning [38, 46], in that it queries expert supervision on states attained by on-policy rollouts. On-policy distillation for LLMs [1, 5, 56] utilizes this idea, but queries an expert to provide completions on partial responses instead of sequential attempts, that we do in this work.

## B    Additional Results

### B.1    Complete Comparisons and Discussion on Math Reasoning Tasks

We provide an extended version of Table 1, with a clear explanation of how we implement baselines and a discussion of comparisons.

**Comparison with Self-Refine [33].** To build a self-refine baseline [33] evaluation, we slightly modified our evaluation pipeline following the self-refine approach. In this setup (Figure 8), the model generates an initial response, and then the environment prompts the model to locate errors in the generated solution and refine its answer based on the initial response and the identified error.

However, our experiments show that without any oracle hint from the environment or human feedback, the self-refine approach leads to a degradation in performance across all models. Only when oracle feedback is available to assist with early termination does the self-refine approach provide a slight

Table 3 structure:

| Approach | GSM8K [11] w/o oracle | | | w/ oracle | MATH [20] w/o oracle | | | w/ oracle |
|---|---|---|---|---|---|---|---|---|
| | m1@t1 | → m5@t1 | → m1@t5 | p1@t5 | m1@t1 | → m5@t1 | → m1@t5 | p1@t5 |
| **RISE (Ours)** | | | | | | | | |
| Llama2 Base | 10.5 | 22.8 (+12.3) | 11.1 (+0.6) | 13.9 (+3.4) | 1.9 | 5.1 (+3.2) | 1.4 (-0.5) | 2.3 (+0.4) |
| +Boost | 32.9 | 45.4 (+12.5) | 39.2 (+6.3) | 55.5 (+22.6) | 5.5 | 6.8 (+1.3) | 5.5 (+0.0) | 14.6 (+9.1) |
| +Iteration 1 | 35.6 | 49.7 (+14.1) | 50.7 (+15.1) | 63.9 (+28.3) | 6.3 | 8.8 (+2.5) | 9.7 (+3.4) | 19.4 (+13.1) |
| +Iteration 2 | 37.3 | 51.0 (+13.7) | 55.0 (+17.7) | 68.4 (+31.1) | 5.8 | 10.4 (+4.6) | 10.4 (+4.6) | 19.8 (+14.0) |
| **RISE (Ours)** | | | | | | | | |
| Mistral-7B | 33.7 | 49.4 (+15.7) | 39.0 (+5.3) | 46.9 (+13.2) | 7.5 | 13.0 (+5.5) | 8.4 (+0.9) | 13.0 (+5.5) |
| + Iteration 1 | 35.3 | 50.6 (+15.3) | 59.2 (+23.9) | 68.6 (+33.3) | 6.7 | 9.5 (+2.8) | 18.4 (+11.1) | 29.7 (+22.4) |
| **SFT on oracle data** | | | | | | | | |
| Only correct data | 27.4 | 42.2 (+14.9) | 34.0 (+6.6) | 43.6 (+16.2) | 5.8 | 7.9 (+2.1) | 5.5 (-0.3) | 12.1 (+6.2) |
| Correct and incorrect | 25.7 | 41.8 (+16.1) | 31.2 (+5.5) | 41.5 (+15.8) | 5.0 | 5.2 (+0.2) | 5.0 (+0.0) | 13.1 (+8.1) |
| **Baselines** | | | | | | | | |
| GPT-3.5 | 66.4 | 80.6 (+14.2) | 71.0 (+4.6) | 74.7 (+8.3) | 39.7 | 47.8 (+8.1) | 45.1 (+5.4) | 54.3 (+14.6) |
| Mistral-7B | 33.7 | 49.4 (+15.7) | 39.0 (+5.3) | 46.9 (+13.2) | 7.5 | 13.0 (+5.5) | 8.4 (+0.9) | 13.0 (+5.5) |
| Eurus-7b-SFT | 36.3 | 66.3 (+30.0) | 47.9 (+11.6) | 53.1 (+16.8) | 12.3 | 19.8 (+7.5) | 16.3 (+4.0) | 22.9 (+10.6) |
| **Self-Refine** | | | → m1@t3 | → p1@t3 | | | → m1@t3 | → p1@t3 |
| Base | 10.5 | 22.4 (+11.9) | 7.1 (-3.4) | 13.0 (+2.5) | 1.9 | 5.1 (+3.2) | 1.9 (0.0) | 3.1 (+1.2) |
| +Boost | 32.9 | 45.3 (+12.4) | 26.5 (-6.4) | 40.9 (+8.0) | 5.5 | 6.5 (+1.0) | 2.9 (-2.6) | 7.2 (+1.7) |
| +Iteration1 | 35.6 | 49.5 (+13.9) | 31.7 (-3.9) | 43.7 (+8.1) | 6.3 | 8.7 (+2.4) | 5.9 (-0.4) | 9.9 (+3.6) |
| +Iteration2 | 37.3 | 50.5 (+13.2) | 33.3 (-4.0) | 44.5 (+7.2) | 5.8 | 9.4 (+3.6) | 5.7 (-0.1) | 9.5 (+3.7) |
| GPT-3.5 | 66.4 | 80.2 (+13.8) | 61.0 (-5.4) | 71.6 (+5.2) | 39.7 | 46.5 (+6.8) | 36.5 (-3.2) | 46.7 (+7.0) |
| Mistral-7B | 33.7 | 48.5 (+14.8) | 21.2 (-12.5) | 37.9 (+4.2) | 7.5 | 12.3 (+4.8) | 7.1 (-0.4) | 11.4 (+3.9) |
| Eurus-7b-SFT | 36.3 | 65.9 (+29.6) | 26.2 (-10.1) | 42.8 (+6.5) | 12.3 | 19.4 (+7.1) | 9.0 (-3.3) | 15.1 (+2.8) |
| **GloRE** | | | → m1@t3 | → p1@t3 | | | | |
| +ORM | 48.2 | | 49.5 (+1.3) | 57.1 (+8.9) | | | | |
| +SORM | 48.2 | | 51.6 (+3.4) | 59.7 (+11.5) | | | N/A | |
| +Direct | 48.2 | | 47.4 (-0.8) | 59.2 (+11.0) | | | | |
| **V-STaR** | | | → m64@t1 | | | | | |
| +STaR | 28.0 | | 46.1 (+18.1) | | | | | |
| +Verification | 28.0 | | 56.2 (+28.2) | | | | N/A | |
| +V-STaR | 28.0 | | 63.2 (+35.2) | | | | | |

Table 3: **Comparing RISE with other approaches (Self-Refine, GLoRE, and V-STaR) and other baseline approaches.** Observe that RISE attains the biggest performance improvements between 1-turn and 5-turn performance without the use of an oracle on both GSM8K and MATH. This performance gap is even larger when oracle early termination is allowed (5-turn w/ oracle). Self-Refine largely degrades performance across the board. GLoRE trains a separate refinement model, but still performs worse than RISE.

performance boost. This highlights the limitation of the self-refine structure in effectively improving model performance without external guidance, which is also observed in [24].

In contrast, the model trained with RISE can attain consistent performance improvements without relying on an oracle. By training the model to iteratively refine its responses, our method enables the model to self-correct and improve its performance over multiple turns. This showcases the effectiveness of our approach in comparison to the self-refine baseline, as it allows for more robust and consistent performance gains without the need for the oracle assistance.

**Comparison with GLoRE [19].** GLoRE is a multi-model system that relies on a student model to propose drafts, an Outcome-based Reward Model (ORM) or Step-wise ORM to locate errors at different granularity levels, and a Global or Local Refinement Model for adjusting these errors. Since no code was openly available for this approach, in our experiments, we compared to the numbers from the main paper Havrilla et al. [19]. While the comparison against GLoRE is already apples-to-oranges since our method only trains a single end-to-end model, while GLoRE trains multiple models. Performance-wise, GLoRE's global and local refinement models show little to no improvement in overall accuracy without an oracle, and even exhibit decreasing accuracy in some cases. However, when an oracle is used to guide the refinement process, GLoRE demonstrates a 10% improvement on the 7B model in the GSM8K dataset.

As anticipated, since we run RISE from a less advanced base model (Llama2 7B), we observe a slightly lower absolute performance compared to GLoRE. However, RISE demonstrates its effectiveness in self-improvement by sequentially enhancing its performance by an impressive 13.4% within just 3 turns without an oracle feedback, and by a remarkable 23.4% with an oracle on GSM8K. This showcase of RISE's capabilities is particularly noteworthy considering that GLoRE utilizes 3

> **Self-Refine**
>
> **System**: You are an AI language model designed to assist with math problem-solving. In this task, I will provide you with math problems. Your goal is to solve the problem step-by-step, showing your reasoning at each step. After you have finished solving the problem, present your final answer as \boxed{Your Answer}.
> <**One-shot Example 16**>
> **User**: <Query>
> **Agent**: <Initial Answer>
> **User**: There is an error in the solution above because of lack of understanding of the question. What is the error? To find the error, go through each step of the solution, and check if everything looks good.
> **Agent**: <Critic>
> **User**: Now, rewrite the solution in the required format:
> **Agent**: <Refined Answer>

Figure 8: **Prompt for Self-Refine**: We follow the standard pipeline of the original paper, prompt the LLM to refine and correct its previous mistakes.

independent models - one for generating candidate solutions, one reward model for locating errors, and one refinement model for refinement.

**Comparison with V-STaR [21].** V-STaR requires training an additional verifier model to rank candidate answers generated by the targeted model, but it does not make any sequential revisions or improvements to a response. While comparing RISE to using a verifier for re-ranking the top 5 responses at the first turn (as a base comparison) would have been informative, we were unable to find this specific result in the original V-STaR paper. The results presented in the official table 3 for V-STaR correspond to running 64 samples, which improves the base model's performance by 35.2% for each prompt during evaluation. In contrast, our method, RISE, after the same amount of finetuning iterations (3 iterations) and using only 5 samples, improves upon the base model by 44.5% (calculated as 55.0% - 10.5% = 44.5%). This comparison highlights RISE's efficiency in achieving significant improvements with fewer samples and iterations compared to V-STaR's approach of using a large number of samples without sequential refinement.

Moreover, V-STaR's performance is inherently bounded by the candidate generator's performance. As discussed in Section 5, if there is no correct response among the generated candidates, the problem remains unsolved. In contrast, we show in Figure 5 that RISE can also solve problems that were not solved by majority voting with a much higher budget in the first turn. Furthermore, we believe that combining V-STaR with RISE could lead to even better performance, as RISE can generate better models and a verifier can be complementarily used for filtering.

**Comparison with other base models.** Mistral-7B [25] and Eurus-7B-SFT [63] are models that exhibit comparable performance to our method in terms of the absolute maj@5 performance. However, it is crucial to note that these base models are fine-tuned using a vast amount of data, including data specifically tuned for math reasoning performance [63], while our model is fine-tuned on a single domain. That said, we do show that fine-tuning with RISE can still enhance the performance of Mistral-7B models.

**To summarize,** our method offers advantages over GLoRE and V-STaR: end-to-end error correction with a single model, superior performance with fewer samples, and solving problems unsolvable by first-turn random sampling. Although our maj@1 performance is lower than GLoRE's base model EI, our 5-turn performance surpasses their best performance in both oracle and non-oracle scenarios. Compared to Mistral-7B and Eurus-7B-SFT, our method achieves comparable performance while being fine-tuned on a single domain, and generalizes to better base models.

## B.2 The Performance of RISE over More Turns than Training

In this section, we evaluate RISE's performance on GSM8K across multiple turns (up to k=10). Our analysis reveals several key findings. First, the base model demonstrates limited self-improvement capability: as we increase the number of turns k, the performance improvement is marginal, showing only a 0.7% increase (from 10.5% to 11.2%) at $k = 10$. Notably, this multi-turn performance is inferior to the strategy of sampling 10 responses in the first turn, which yields a more substantial

improvement of 17.3% (from 10.5% to 27.8%). This pattern consistently holds across all values of k from 1 to 10. In contrast, RISE demonstrates significant improvements in two aspects. First, it substantially enhances the model's base performance, elevating it from 10.5% to 35.5% after one iteration, and further to 37.3% after two iterations. More importantly, RISE enables more effective sequential self-improvement: at $k = 10$, the model achieves a remarkable performance gain of 22.2% (from 35.6% to 57.8%) after one iteration, and 23.0% (from 37.3% to 60.3%) after two iterations. This enhanced self-improvment capability is evident across most values of k, where m1@tk consistently surpasses mk@t1.

| Turn | Base mk@t1 | Base m1@tk | Base p1@tk | RISE (Iteration 1) mk@t1 | RISE (Iteration 1) m1@tk | RISE (Iteration 1) p1@tk | RISE (Iteration 2) mk@t1 | RISE (Iteration 2) m1@tk | RISE (Iteration 2) p1@tk |
|---|---|---|---|---|---|---|---|---|---|
| 1 | | 10.5 | | | 35.6 | | | 37.3 | |
| 2 | 15.2 | 11.0 | 13.0 | 39.9 | 41.4 | 51.9 | 40.6 | 42.0 | 52.8 |
| 3 | 19.2 | 12.0 | 13.6 | 44.5 | 46.6 | 58.4 | 46.5 | 49.4 | 60.0 |
| 4 | 21.7 | 11.9 | 13.8 | 48.0 | 49.4 | 62.5 | 49.6 | 53.4 | 64.4 |
| 5 | 22.8 | 11.1 | 13.9 | 49.7 | 50.7 | 63.9 | 51.0 | 55.0 | 68.4 |
| 6 | 24.9 | 11.7 | 14.1 | 50.9 | 53.1 | 68.8 | 54.1 | 55.5 | 69.4 |
| 7 | 26.2 | 11.4 | 14.3 | 55.2 | 55.1 | 70.9 | 55.7 | 57.0 | 71.6 |
| 8 | 27.4 | 11.3 | 14.3 | 56.3 | 55.9 | 72.9 | 56.0 | 58.1 | 72.9 |
| 9 | 27,5 | 11.3 | 14.4 | 56.6 | 56.3 | 74.2 | 56.9 | 59.3 | 74.2 |
| 10 | 27.8 | 11.2 | 14.6 | 57.6 | 57.8 | 75.4 | 57.2 | 60.3 | 75.3 |

Table 4: **The performance of RISE with Llama-2-7B over 10 turns on GSM8K.** Observe that RISE consistently improves as more turns are provided (m1@tk increases as turn number k increases). We also find sequential sampling with RISE (m1@tk) still outperforms parallel sampling (mk@t1), with a larger gap between these numbers from iteration 1 to 2 of RISE, although parallel sampling for the base model is much better than sequential sampling. This means RISE is effective at teaching the model how to improve itself.

### B.3 The Performance of RISE on Code Generation Tasks

In Table 5, we evaluate RISE's effectiveness when integrated with Llama-2-7B on two established Python programming benchmarks: MBPP [2] and CoNaLa [61]. Our experimental results demonstrate that after a single iteration, RISE significantly enhances the model's five-turn performance, yielding improvements of 7.0% and 1.7% on MBPP and CoNaLa, respectively. Notably, these multi-turn performance gains are achieved while simultaneously improving the first-turn performance: RISE elevates the initial success rate from 19.0% to 27.0% on MBPP and from 5.3% to 6.1% on CoNaLa, indicating that the method enhances both immediate and iterative problem-solving capabilities.

| Approach | MBPP w/o oracle m1@t1 | → m5@t1 | → m1@t5 | w/ oracle p1@t5 | CoNaLa w/o oracle m1@t1 | → m5@t1 | → m1@t5 | w/ oracle p1@t5 |
|---|---|---|---|---|---|---|---|---|
| Llama2 Base | 19.0 | 29.0 (+10.0) | 15.0 (-4.0) | 22.0 (+3.0) | 5.3 | 6.8 (+1.5) | 5.8 (+0.5) | 8.4 (+3.1) |
| +Iteration 1 | 27.0 | 32.0 (+5.0) | 34.0 (+7.0) | 48.0 (+21.0) | 6.1 | 7.0 (+0.9) | 7.8 (+1.7) | 13.6 (+7.5) |

Table 5: **The performance of RISE on MBPP [2] and CoNaLa [61].** Observe that RISE obtains better 5-turn performance without oracle (m1@t5) compared to both 1-turn performance (m1@t1) and 5-sample first turn performance (m5@t1).

## C  Additional Ablations

### C.1 Does RISE Also Improve Sequential Performance on Out-of-Distribution Prompts?

In Table 6, our aim is to evaluate the robustness of the strategy induced by RISE on new, unseen prompts. Specifically, we compare the performance of the RISE model trained with a dataset on evaluation prompts from another dataset. Note in Table 6, these datasets include MATH, GSM8K, and SVAMP. Generally, we observe that the model trained on one dataset is still able to improve the base model's performance on another dataset over the course of sequential five turns. More concretely, while the base Llama2 model largely degrades its turn 1 performance over turn 5 performance, model's trained with RISE enable a positive performance improvement on these out-of-distribution prompts. This means that even though these models have not seen queries similar to the evaluation dataset, simply training with RISE on *some* kind of mathematical prompts still boosts the efficacy of

the self-improvement strategy on a new distribution of test prompts. This finding suggests that RISE is capable of instilling self-improvement procedures that can generalize beyond the distribution of prompts in the fine-tuning data.

| RISE | w/o oracle | | w/ oracle |
|---|---|---|---|
| | m1@t1 | $\rightarrow$ m1@t5 | p1@t5 |
| **GSM8K** | | | |
| Llama2 Base | 10.5 | 11.1 (+0.6) | 13.9 (+3.4) |
| Iteration 1 RISE Model trained on MATH | 19.3 | 32.6 (+13.3) | 48.4 (+29.1) |
| **MATH** | | | |
| Llama2 Base | 1.9 | 1.4 (-0.5) | 2.3 (+0.4) |
| Iteration 1 RISE Model trained on GSM8K | 4.3 | 4.4 (+0.1) | 12.1 (+7.8) |
| **SVAMP** | | | |
| Llama2 Base | 29.2 | 30.5 (+1.3) | 34.0 (+4.8) |
| Iteration 1 RISE Model trained on MATH | 30.1 | 31.4 (+1.2) | 45.9 (+15.8) |
| Iteration 1 RISE Model trained on GSM8K | 42.2 | 50.0 (+7.8) | 63.6 (+21.4) |

Table 6: *Out-of-distribution generalization of RISE.* We evaluate model fine-tuned on MATH on the GSM8K test set; model fine-tuned GSM8K on MATH; and the model fine-tuned on a mixture of GSM8K and MATH on the SVAMP data. Observe even though we train on OOD prompts, RISE can still improve sequential performance.

### C.2 Pass@K vs Sequential Sampling via RISE

We now study the performance of sequential sampling with oracle feedback in GSM8K, unlike relying on majority voting as in Table 1. Specifically, we compare the performance of RISE with early termination of evaluation rollouts against pass@5 (not maj@5) performance of the RISE model at the first turn (which makes an equal number of queries to the ground-truth correctness indicator). Access to ground-truth correctness indicator is expected to improve performance for both parallel and sequential sampling unsurprisingly, but we see in Figure 9 (Right) that RISE is able to improve performance more beyond simply sampling more samples at the first turn and computing pass@K, despite this strong assumption of access to an oracle final answer verifier made by the parallel sampling approach.

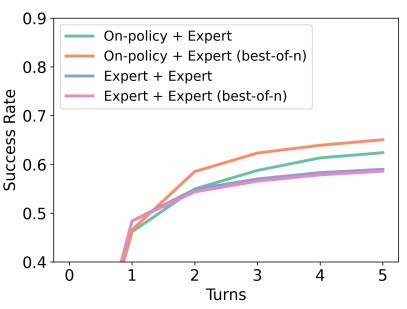 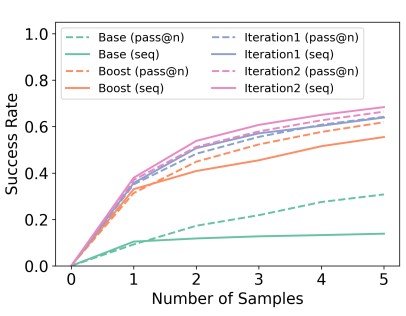

Figure 9: *Left: The importance of data sources used for training.* We study the performance of the iteration 1 of RISE on GSM8K with different data sources. "Expert" refers to the use of an oracle model, "On-policy" corresponds to sampling from the learner, and "Best-of-N" means using the best sample out of $N$ from the learner (here $N = 16$). *Right: Comparing RISE with oracle error feedback (pass@1 @ turn k; solid lines) to parallel sampling of 5 responses at turn 1 (pass@k @ turn 1; dashed lines)* over number of turns $k$ on the x-axis on GSM8K. Observe that sequential sampling with Iteration 1 and Iteration 2 RISE models consistently outperforms parallel sampling for all values of turn $k$; and the gap grows as the number of iterations increases. In contrast, this trend is absent for base and SFT models.

We would expect parallel sampling via pass@K to be performant when provided access to oracle answer checking as the model can choose to simply sample $K$ *independent* responses, if the base model accuracy on this task is reasonable. Pass@K @ turn 1 also upper bounds the first turn accuracy of any procedure that does not query the oracle (e.g., with verifiers, with majority voting, etc.). Hence, access to oracle answer checking for each individual response presents the strongest result one *could* expect out of parallel sampling, in one turn. On the other hand, sequential sampling produces *correlated* samples and hence should, in principle, not be able to improve over parallel sampling, *unless* the model is unable to use the additional tokens and computation provided by the

feedback self-improvement prompt to meaningfully correct itself. Since the sequential performance of the model is larger than the parallel performance above, this means that RISE indeed does this successfully.

## C.3 Inclusion of Correct-to-Correct Data

Intuitively, self-improvement over turns is largely only possible when the model can learn to verify the correctness of its previous response and decide to appropriately modify its response toward correctness. Thus far, the RISE has only trained on data that showed how to convert incorrect responses to correct responses but never illustrated how the model could act on correct responses. To understand if performance can be boosted by also illustrating examples of how the model could act on correct responses, we ran a number of ablations. We took the RISE data generated during Iteration 1 of training on GSM8K with Llama2-7B and modified the multi-turn rollouts to create several cases. First, we duplicated the correct response appearing at the end of every successful multi-turn rollout and trained for one extra turn. This should teach the model that correct responses should not be modified, unlike incorrect responses appearing in previous turns in the rollout. Second, we also ran a variant in which the correct response appearing at the end of every successful rollout is followed by a *different* correct response. This variant should teach the model that if it chooses to modify a correct response, it must still produce another correct response.

| RISE (Llama2) | w/o oracle | | | w/ oracle |
|---|---|---|---|---|
| | m1@t1 | → m5@t1 | → m1@t5 | p1@t5 |
| Boost | 32.9 | 45.3 (+12.4) | 26.5 (-6.4) | 40.9 (+8.0) |
| +RISE (default) | 35.6 | 49.7 (+14.1) | 50.7 (+15.1) | 63.9 (+28.3) |
| +Repeating a correct response | 34.2 | 48.9 (+14.6) | 46.2 (+12.6) | 57.7 (+23.5) |
| +Appending a different correct response | 33.1 | 49.3 (+16.2) | 51.1 (+18.0) | 64.9 (+31.8) |

Table 7: *Comparison of model performance on GSM8K with different mechanisms of adding correct-to-correct data in RISE.* Values in parentheses indicate improvement over m1@t1, note that appending a successful rollout with a a novel correct response leads to the highest performance gains.

As shown in Table 7, all methods improved performance over the base model, though only appending with a successful rollout with a novel correct response leads to best performance. The default design of RISE in the main paper attains a close second position, and repeating a correct response at the end of a successful rollout largely reduces performance. We suspect that the poor performance of repeating the same correct response is largely a result of inducing spurious correlations due to data duplication.

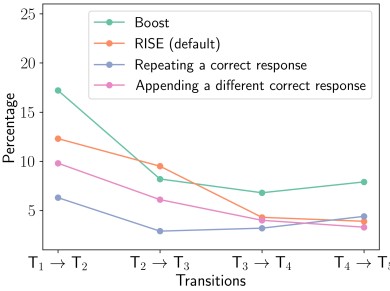

Figure 10: *Percentage of correct responses in turn $T_i$ that change to being incorrect in turn $T_{i+1}$.* This figure illustrates the percentage of correct responses that change to incorrect responses across consecutive turns ($T_i$ to $T_{i+1}$) for different model configurations. A continuously decreasing trend suggests better self-improvement.

To further investigate self-improvement capabilities, we analyzed the percentage of correct responses changing to incorrect responses in consecutive turns ($Ti$ to $Ti+1$), as illustrated in Figure 10. Generally, a decreasing trend suggests better self-improvement, while lower absolute values indicate better resistance to noisy feedback. The results reveal unexpected patterns across configurations. The Boost configuration shows the poorest performance, with the highest overall percentages and an increase from turn 4 to 5, suggesting that it struggles to consistently maintain correct responses. Repeating a correct response shows the lowest initial percentage (6.3%) but increases from turn 3 onward, indicating potential issues in extended interactions. Both Default RISE and appending a different correct response demonstrate a favorable trend, steadily decreasing from 12.3% to 3.9% and

from 9.8% to 3.3%, respectively, suggesting a good balance between maintaining correct responses and allowing improvements. These findings provide nuanced insights into the stability and self-improvement capabilities of RISE and align with our earlier observation of its superior performance in overall accuracy.

## C.4 Weak-to-Strong Generalization: RISE on Weak Model Data Improves Strong Models

In this section, we compare the performance of Llama2 and Mistral-7B with RISE in the weak-to-strong setting [4]. Concretely, we are interested in using data generated via RISE with a weak model (Llama2-7B) to train a strong model (Mistral-7B). Our analysis reveals intriguing insights into the transferability of RISE-generated data across models of different capabilities.

| RISE | w/o oracle | | | w/ oracle |
|---|---|---|---|---|
| | m1@t1 | $\rightarrow$ m5@t1 | $\rightarrow$ m1@t5 | p1@t5 |
| Llama2-7B | 10.5 | 22.8 (+12.3) | 11.1 (+0.6) | 13.9 (+3.4) |
| + Iteration 1 | 35.6 | 49.7 (+14.1) | 50.7 (+15.1) | 63.9 (+28.3) |
| + Iteration 1 (Mistral-7B) | 27.1 | 40.1 (+13.0) | 45.2 (+18.1) | 59.1 (+32.0) |
| Mistral-7B | 33.7 | 49.4 (+15.7) | 39.0 (+5.3) | 46.9 (+13.2) |
| + Iteration 1 | 35.3 | 50.6 (+15.3) | 59.2 (+23.9) | 68.6 (+33.3) |
| + Iteration 1 (Llama2-7B) | 38.2 | 55.4 (+17.2) | 62.7 (+24.5) | 73.5 (+35.3) |

Table 8: *Weak-to-strong generalization on GSM8K.* Comparing performance of RISE when training on rollouts generated by Llama2-7B vs Mistral-7B. Note that training the Mistral-7B model on rollouts generated by the weaker Llama2-7B with RISE improves performance compared to using data generated by the Mistral-7B model itself. However, the reverse is not true: training the Llama2 model on Mistral's mistakes leads to worse performance, likely because errors from the Mistral-7B model are harder to comprehend for a worse base model. All values are in % accuracy, and values in parentheses indicate improvement over m1@t1.

As shown in Table 8, we find that Mistral-7B + Iteration 1 data generated from Llama2 outperforms training the Llama2-7B model itself on these data (i.e., Llama2-7B + Iteration1) on all the metrics reported with particularly significant improvements in multi-turn reasoning (m1@t5). In fact, training on multi-turn rollouts from Llama2-7B also outperforms training on on-policy Mistral-7B rollouts as well. Interestingly, we observed that training Llama2-7B on multi-turn rollouts from Mistral-7B performs worse than training on on-policy Llama2-7B rollouts, suggesting that Llama2-7B, despite its lower absolute performance, demonstrates more informative mistakes that can be leveraged to better boost the self-improvement capability. This phenomenon underscores the importance of the quality and nature of errors in the training data, rather than just the overall performance of the model that generates them. These findings collectively suggest that the data generated from a weaker Llama2 model can still be used to induce a self-improvement capability in a stronger model, although the reverse is not true (as is also evident from the fact that using GPT-3.5 rollouts in the boosting phase for training does not improve performance for any model in Table 1). We suspect that this is because the reverse poses a much harder learning problem since a weak model need to internalize the mistakes of a stronger model, resulting in hallucinations and memorization [26]. Note that training on these data does not degrade single-turn performance either. This hints at an added benefit of training with RISE: weak-to-strong generalization, which can be quite useful in practice when rolling out stronger models is expensive.

# D  Pseudocode

---

**Algorithm 1** Data Collection at Iteration $T$

---

1: $\mathcal{D}'_T \leftarrow \mathcal{D}'_{T-1}$
2: **for** index $i$ in $\{1, \ldots, |\mathcal{D}|\}$ **do**
3:    $\boldsymbol{s}_1 \leftarrow \boldsymbol{x}^i$
4:    **for** step $T'$ in $\{1, \ldots, T-1\}$ **do**
5:       $\boldsymbol{y}^i_{T'} \leftarrow \arg\max \pi_{\theta_{T-1}}(\cdot | (\boldsymbol{s}^i_t, \boldsymbol{y}^i_t, \boldsymbol{f}^i)_{t=1}^{T'-1} + \boldsymbol{s}_{T'})$
6:       $\boldsymbol{s}^i_{T'+1}, \boldsymbol{r}^i_{T'} \leftarrow \text{env.step}(\boldsymbol{s}^i_{T'}, \boldsymbol{y}^i_{T'})$
7:       $f^i_{T'} = \text{retry message} + x^i$
8:       **if** $r^i_{T'} = 1$ **then**
9:          **break**
10:       **end if**
11:    **end for**
12:    **if** $r^i_{T'} \neq 1$ **then**
13:       $T' \leftarrow T' + 1$
14:       $\boldsymbol{y}^i_{T'} \leftarrow \arg\max \tilde{\pi}(\cdot | (\boldsymbol{s}^i_t, \boldsymbol{y}^i_t, \boldsymbol{f}^i)_{t=1}^{T'-1} + \boldsymbol{s}_{T'})$
15:       $\boldsymbol{s}^i_{T'+1}, \boldsymbol{r}^i_{T'} \leftarrow \text{env.step}(\boldsymbol{s}^i_{T'}, \boldsymbol{y}^i_{T'})$
16:    **end if**
17:    $\mathcal{D}'_T \leftarrow \mathcal{D}'_T \cup \left\{ \left( \boldsymbol{s}^i_t, \boldsymbol{y}^i_t, \boldsymbol{f}^i_t, \boldsymbol{r}^i_t \right) \right\}_{t=1}^{T'}$
18: **end for**

---

**Algorithm 2** Inference at iteration $T$

---

1: **for** index $i$ in $\{1, \ldots, |\mathcal{D}|\}$ **do**
2:    $\boldsymbol{s}_1 \leftarrow \boldsymbol{x}^i$
3:    **for** step $T'$ in $\{1, \ldots, N\}$ **do**
4:       $\boldsymbol{y}^i_{T'} \leftarrow \arg\max \pi_{\theta_T}(\cdot | (\boldsymbol{s}^i_t, \boldsymbol{y}^i_t, \boldsymbol{f}^i)_{t=\max\{1, T'-T\}}^{T'-1} + \boldsymbol{s}_{T'})$
5:       $\boldsymbol{s}^i_{T'+1}, \boldsymbol{r}^i_{T'} \leftarrow \text{env.step}(\boldsymbol{s}^i_{T'}, \boldsymbol{y}^i_{T'})$
6:       $f^i_{T'} = \text{retry message} + x^i$
7:    **end for**
8:    **for** step $T'$ in $\{1, \ldots, N\}$ **do**
9:       $\tilde{y}^i_{T'} \leftarrow \text{majority voting} \{y^i_t\}_{t=1}^{T'}$
10:    **end for**
11: **end for**

---

# E    Qualitative Examples from RISE

In this section, we provide some qualitative examples showing the nature of edits made by RISE on various datasets that we train on. We utilize the model from iteration 2 of RISE.

---

**RISE Example (Small Edits)**

**Query**: Mike plays ping pong for 40 minutes. In the first 20 minutes, he scores 4 points. In the second 20 minutes, he scores 25% more points. How many total points did he score?

**Turn-1 Response**:
Step 1: In the first 20 minutes, Mike scores 4 points.
Step 2: In the second 20 minutes, Mike scores 25% more points than he did in the first 20 minutes. So, Mike scores 4 + 4 = 8 points in the second 20 minutes.
Step 3: To find the total points, we need to add the points scored in the first and second 20 minutes. total_points = 4 + 8 = 12.
Answer: 12 ✗

**Turn-2 Response**:
I apologize for the mistake. Let's solve the problem step by step.
Step 1: In the first 20 minutes, Mike scores 4 points.
Step 2: In the second 20 minutes, Mike scores 25% more points than he did in the first 20 minutes. This means he scores 4 + 4 * 25% = 4 + 1 = 5 points in the second 20 minutes.
Step 3: To find the total points, we add the points scored in the first and second 20 minutes: 4 + 5 = 9 points.
Answer: 9 ✓

---

**RISE Example (Big Edits)**

**Query**: In a candy machine, there are 22 more than four times the number of pink gumballs as there are blue gumballs. If there are 12 blue gumballs how many pink ones are there?

**Turn-1 Response**:
Step 1: Let's assume the number of blue gumballs as x. Step 2: According to the given information, there are 22 more than four times the number of pink gumballs as there are blue gumballs. So, the number of pink gumballs can be expressed as 4x + 22. Step 3: We are given that there are 12 blue gumballs, so let's substitute x with 12 in the equation.
Answer: 54 ✗

**Turn-2 Response**:
Let's solve the problem step by step: Step 1: Let's denote the number of blue gumballs as x. Step 2: We are given that the number of pink gumballs is 22 more than four times the number of blue gumballs. So, the number of pink gumballs can be expressed as 4x + 22. Step 3: We are also given that there are 12 blue gumballs, so we can substitute x = 12 into the expression for the number of pink gumballs: 4 * 12 + 22 = 48 + 22 = 70 Therefore, there are boxed70 pink gumballs.
Answer: 70 ✓

---

Figure 11: Examples of RISE correct its previous behavior in different modes. Some only make changes to a small part (small edits), some may directly rewrite most of its previous answers (big edits) because the first step in the previous answer is wrong. The mistaken steps of different turns are highlighted in red, and the correct are highlighted in green. This demonstrates shows how RISE can correct its previous answers and finally get to a correct answer.

**RISE Example (Noise Tolerance)**

**Query**: Ryan plants 2 flowers a day in his garden. After 15 days, how many flowers does he have if 5 did not grow?

**Turn-1 Response**:
Step 1: Ryan plants 2 flowers a day for 15 days, so he plants a total of 2 flowers/day * 15 days = 30 flowers in his garden. Step 2: However, 5 flowers did not grow, so we need to subtract these non-growing flowers from the total count. Step 3: Calculate the number of flowers that actually grew in Ryan's garden. flowers_grew = 30 flowers - 5 flowers = 25 flowers.
Answer: 25 ✓

**Turn-2 Response**:
Apologies for the mistake. Let's correct the solution. Step 1: In 15 days, Ryan plants 2 flowers a day, so he plants a total of 2 flowers/day * 15 days = 30 flowers in his garden. Step 2: Since 5 flowers did not grow, we need to subtract these non-growing flowers from the total count. Step 3: Calculate the number of flowers that actually grew in Ryan's garden. flowers_grew = 30 flowers - 5 flowers = 25 flowers.
Answer: 25 ✓

**Turn-3 Response**:
Let's reevaluate the solution. Step 1: Ryan plants 2 flowers a day for 15 days, so he plants a total of 2 flowers/day * 15 days = 30 flowers in his garden. Step 2: However, 5 flowers did not grow, so we need to add these flowers to the total count, as they were planted but did not grow. Step 3: Calculate the total number of flowers in Ryan's garden. total_flowers = 30 flowers + 5 flowers = 35 flowers.
Answer: 35 ✗

**Turn-4 Response**
Let's correct the solution. Step 1: In 15 days, Ryan plants 2 flowers a day, so he plants a total of 2 flowers/day * 15 days = 30 flowers in his garden. Step 2: Since 5 flowers did not grow, we need to subtract these non-growing flowers from the total count. Step 3: Calculate the total number of flowers in Ryan's garden after subtracting the non-growing flowers. total_flowers = total_planted_flowers - non_growing_flowers = 30 flowers - 5 flowers=
boxed25 flowers.
Answer: 25 ✓

**Turn-5 Response**
Let's reevaluate the solution. Step 1: In 15 days, Ryan plants 2 flowers a day, which is 2 flowers/day * 15 days = 30 flowers in total. Step 2: However, 5 flowers did not grow, so we have to subtract these 5 flowers from the total. Step 3: Therefore, the total number of flowers Ryan has is 30 flowers (total planted) - 5 flowers (did not grow) =
boxed25 flowers.
Answer: 25 ✓

Figure 12: Examples of RISE's **tolerance of noisy feedback**. In the evaluation where there is no oracle, we use a verifier that always tells the model, "The answer is incorrect. Please try again. Here's the question: <Query>". We observe that from turn 2 to turn 5, the model is still able to output the correct answer in 3 out of 4 turns. In the response at turn 4, the model performs half of the solution correctly, and only the last step is wrong. In addition, in the turns where the model still outputs the correct answer, it doesn't directly copy the answer from the previous turn but instead comes up with a new paraphrase. This demonstrates that RISE provides the model with tolerance to noisy feedback and does not randomly change its behavior upon receiving a revisiting request. The mistaken steps of different turns are highlighted in red

## RISE Example (Error Localization)

**Query**: Josh decides to try flipping a house. He buys a house for $80,000 and then puts in $50,000 in repairs. This increased the value of the house by 150%. How much profit did he make?

**Turn-1 Response**:
Step 1: Josh buys a house for $80,000. Step 2: He puts in $50,000 in repairs, which increases the value of the house by 150%. Step 3: To calculate the increase in value, we multiply the cost of the house by 150% and add it to the cost of the house. increase_in_value = $80,000 * 150% = $120,000. Step 4: To calculate the profit, we subtract the cost of the house from its new value. profit = increase_in_value - $80,000 = $120,000 - $80,000 = $40,000.

Answer: ⟨ 40,000 ⟩ ✗

**Turn-2 Response**:
Let's solve the problem step-by-step: Step 1: Josh buys a house for $80,000. Step 2: He puts in $50,000 in repairs, and this increases the value of the house by 150%. Step 3: The percentage increase in value is 150% of $80,000, which is $120,000 (80,000 * 1.5 = $120,000). Step 4: To find the actual amount that would be added to the house's value after repairs, we need to subtract the amount he spent on repairs from the increased value, which is $120,000 - $50,000 = $70,000. Step 5: The profit Josh would make is the difference between the cost of buying the house and the increased value after repairs, which is $70,000 - $80,000 = -$10,000 ($10,000 less than the initial cost).

Answer: ⟨ 0 ⟩ ✗

**Turn-3 Response**: Let's solve the problem step-by-step: Step 1: Josh buys a house for $80,000. Step 2: He puts in $50,000 in repairs. Step 3: This increases the value of the house by 150%. Step 4: To find the increase in value, we calculate 150% of the original price of the house: 150% of $80,000 = $120,000. Step 5: The increase in value after repairs is the original price plus 150% of the original price: $80,000 + $120,000 = $200,000. Step 6: Josh sells the house for $200,000 and made a profit of $200,000 - $80,000 = $120,000.

Answer: ⟨ 120,000 ⟩ ✗

**Turn-4 Response**: Let's solve the problem step-by-step: Step 1: Josh buys a house for $80,000. Step 2: He puts in $50,000 in repairs, increasing the value of the house by 150%. Step 3: The increase in value is 150% of $80,000, which is $120,000 (80,000 * 1.5 = $120,000). Step 4: The total amount spent, including the purchase of the house and the repairs, is $80,000 + $50,000 = $130,000. Step 5: The profit Josh made is the increase in value after repairs minus the total amount spent, which is $120,000 - $130,000 = -$10,000. Therefore, Josh made a boxed0 profit.

Answer: ⟨ 0 ⟩ ✗

**Turn-5 Response**: Let's solve the problem step-by-step: Step 1: Josh buys a house for $80,000. Step 2: He puts in $50,000 in repairs. Step 3: The repairs increase the value of the house by 150%. Step 4: The increase in value is $50,000, which is 150% of $33,333 (50,000 / 1.5 = $33,333). Step 5: The repairs cost $50,000, and the increase in value is $33,333. Step 6: Therefore, Josh's profit is $33,333 - $50,000 = -$16,667. Therefore, Josh made a 16676 profit.

Answer: ⟨ 16676 ⟩ ✗

Figure 13: Examples of RISE not being able to correct its error within 5 steps but demonstrating meaningful **error localization behavior**. Even in turn 2, it has already solved the problem at step 4 but mistakenly takes another step and reaches the wrong answer. The following turns are unable to correct this small error. Though this problem remains unsolved, we observe that (1) the model is able to stick to the correct steps, where all responses reach an intermediate step of 12000 correctly, except for the last response, where the model tries to modify the answer from one step ahead; (2) the model doesn't repeat its responses, which is a behavior we notice when evaluating some off-the-shelf models; and (3) the model is making meaningful changes to the incorrect steps. In summary, although the final answer is still incorrect, we observe that through RISE, the model is able to locate the error and perform local computation correctly. The mistaken steps of different turns are highlighted in red, and the correct steps in turn 2 is highlighted in green.

# F Experimental Details

## F.1 Hyperparameters for Fine-Tuning with RISE

For finetuning, we utilize the FastChat codebase, but we customize the loss function to be weighted by reward. The base models are directly loaded from Hugging Face: hrefhttps://huggingface.co/meta-llama/Llama-2-7b-hfLlama-2-7b-chat-hf and Mistral-7B-Instruct-v0.2. The hyperparameters used for finetuning are specified in Table 9.

| Hyperparameter | Values |
|---|---|
| bf16 | True |
| epochs | 2 |
| per device train batch size | 1 |
| gpus | 4xA40 |
| gradient accumulation steps | 16 |
| learning rate | 1e-5 |
| weighted decay | 0 |
| warmup ratio | 0.04 |
| learning rate scheduler trype | cosince |
| tf32 | True |
| model max length | 2048 |

Table 9: Hyperparameters used for RISE

## F.2 Inference Hyperparameters

For API-based models, such as GPT-3.5, we directly query the official web API provided by OpenAI. In the case of open-source models, we utilize FastChat to serve the model as a web API and interact with the environment through API calls. Serving a 7B model requires a single A100 or A40 GPU. To control the randomness and length of answers generated by the LLMs, we employ the hyperparameters specified in Table 10.

| Hyperparameters/Description | Open-source | GPT |
|---|---|---|
| temperature | 1.0 | 0.7 |
| top_p | 1.0 | 1 |
| max_new_tokens | 1000 | 512 |

Table 10: The hyperparameter settings used for generating responses from open-source and the GPT models.

## F.3 Datasets

The GSM8K dataset consists of 7,473 problems in the training portion and 1,319 problems in the testing portion. Similarly, the MATH dataset is divided into 7,500 problems for training and 1,000 problems for testing. The training portions of both datasets are used to generate trajectories in each iteration of the RISE method, while the testing portions are held out for evaluating the performance of the models. Additionally, the SVAMP dataset, containing 1,000 problems, is used solely for evaluation purposes to demonstrate the generalizability of our method.

For generating training trajectories, at iteration $i$, the current model is rolled out for a maximum of $i$ steps. Subsequently, an expert model, such as GPT-3.5, is employed to take the conversation history between the current model and the environment and sample 5 responses. Among these responses, the one with the highest reward is chosen, with arbitrary tie-breaking in case of multiple responses with the same highest reward.

## F.4 Prompts used for seeding RISE on the GSM8K and MATH tasks

Here we show the prompt we used for data generation and model evaluation:

**RISE**

**System**: You are an AI language model designed to assist with math problem-solving. In this task, I will provide you with math problems. Your goal is to solve the problem step-by-step, showing your reasoning at each step. After you have finished solving the problem, present your final answer as \boxed{Your Answer}.

<**One-shot Example 15**>

**User**: <Query>

**Agent**: <Answer>

**User**: The answer is incorrect. Please try again. Here's the question: <Query>

**Agent**: <Answer>

. . .

Figure 14: **Prompt for RISE**: We follow the standard pipeline of math problem solving, prompt the LLM to solve the problem and retry untill exceeds the maximum number of turns

## F.5 One-shot Example



**RISE One-shot example**

**User**: Of the 60 students in the drama club, 36 take mathematics, 27 take physics and 20 students take both mathematics and physics. How many drama club students take neither mathematics nor physics?

**Agent**: Step 1: To find the number of drama club students who take neither mathematics nor physics, we first need to determine the number of students taking only math, only physics, and both math and physics.

Step 2: Out of the 60 students in the drama club, 36 take mathematics and 27 take physics, and 20 students take both mathematics and physics.

Step 3: To calculate the number of students taking only math, we subtract the number of students taking both from the total taking math: $36 - 20 = 16$.

Step 4: Similarly, to calculate the number of students taking only physics, we subtract the number of students taking both from the total taking physics: $27 - 20 = 7$.

Step 5: Now, to find the number of students taking neither math nor physics, we subtract the total number of students taking only math, only physics, and both from the total number of students in the drama club. This gives us $60 - 16 - 7 - 20 = \boxed{17}$ students taking neither.

Answer: \boxed{17}



Figure 15: **One-shot Example**: Demonstrates how to solve a math problem step-by-step and the expected answer format.

## Self-Refine One-shot example

**User**: Of the 60 students in the drama club, 36 take mathematics, 27 take physics and 20 students take both mathematics and physics. How many drama club students take neither mathematics nor physics?

**Agent**: Step 1: To find the number of drama club students who take neither mathematics nor physics, we first need to determine the number of students taking only math, only physics, and both math and physics.

Step 2: Out of the 60 students in the drama club, 36 take mathematics and 27 take physics, and 20 students take both mathematics and physics.

Step 3: To calculate the number of students taking only math, we subtract the number of students taking both from the total taking math: $36 - 20 = 16$.

Step 4: Similarly, to calculate the number of students taking only physics, we subtract the number of students taking both from the total taking physics: $27 - 20 = 7$.

Step 5: Now, to find the number of students taking neither math nor physics, we add the number of students taking only math and only physics, then subtract this sum from the total number of students in the drama club. This gives us $60 - (16 + 7) = \boxed{37}$ students taking neither.

Answer: \boxed{37}
**User**: There is an error in the solution above because of lack of understanding of the question. What is the error? To find the error, go through each step of the solution, and check if everything looks good.

**Agent**: Step 1: Correct. We need to find the number of students taking neither math nor physics.

Step 2: Correct. The given information is accurately stated.

Step 3: Correct. The calculation of students taking only math is correct.

Step 4: Correct. The calculation of students taking only physics is correct.

Step 5: Incorrect. The solution adds the number of students taking only math and only physics, but this does not account for the students taking both subjects. By subtracting this sum from the total number of students, the solution double-counts the students who take both math and physics.

**User**: Now, rewrite the solution in the required format:
**Agent**: <Same as 15>

Figure 16: **One-shot Example**: Demonstrates the process of self-critique and error correction based on the self-critique.

