# OpenReview forum: "Recursive Introspection: Teaching Language Model Agents How to Self-Improve"
_NeurIPS.cc/2024/Conference — NeurIPS 2024 poster_

### Official Review · Reviewer_BtCg · 2024-07-04

**Soundness:** 3
**Presentation:** 4
**Contribution:** 4
**Rating:** 7
**Confidence:** 4

**Summary:**

This paper proposes a general paradigm for fine-tuning LLMs such that the LLM can iteratively refine from its previous outputs in an in-context learning fashion. Experiments are conducted on math tasks where the learned model can self-improve with multi-turn outputs.

**Strengths:**

1. The proposed RISE framework is novel and general. The overall framework can be possibly extended to a wide spectrum of tasks. Although the current form of the method remains simple, it can serve as a good starting point for learning self-adaptive LLMs.

2. The experiments are solid with a diverse collection of benchmarks as well as baselines including GLORE and Self-Refine. The ablation studies and analysis are thorough.

3. The paper is well-written and easy to follow.

**Weaknesses:**

There are no major weaknesses in this paper. But there may be some potential directions for further improvement.

1. The current learning method fundamentally remains a distillation process, either from a teacher or from the base model itself. Is it possible to directly perform RL over the trajectories? Note that this is essentially an in-context RL process. It would be interesting to see whether this can be achieved.

2. The experiments are primarily conducted on Math benchmarks. The paper can be even stronger if other tasks can be considered.

3. The authors list two possible reasons why having an improving trajectory can lead to better final outputs. I fully agree with the two reasons. It can be more convincing if some concrete examples supporting these two hypotheses can be discussed.

4. RISE looks so much like in-context RL[1,2] because in-context RL also aims to learn a model that you can put trajectories in the context window and then the LLM can run RL to improve the output. Some discussions on this can be appreciated.

[1] In-context Reinforcement Learning with Algorithm Distillation, M. Laskin et al., https://arxiv.org/abs/2210.14215

[2] Supervised Pretraining Can Learn In-Context Reinforcement Learning, J. Lee et al., https://arxiv.org/abs/2306.14892

**Questions:**

See comments above.

**Limitations:**

Limitations have been well addressed in the paper.

---

> ### Author Rebuttal · Authors · 2024-08-07
>
> Thanks for your feedback and for a positive assessment of our work. We are glad that you think there is no major weakness and we appreciate the suggestions for future improvement. We fully agree that the literature on in-context RL is quite related, we discuss below and will add a discussion. We provide **new results on MBPP and CoNaLa (coding benchmarks) and find RISE to outperform single-turn training or parallel sampling at the first turn (Table 3)**. We also add new results to justify why training on a multi-turn rollout leads to more flexible model capacity (Figure 1 and Table 2). **Please let us know if your concerns are addressed and we would appreciate it if you might be willing to upgrade your score.**
> ___
>
> ## **[New result]** Other benchmarks
>
> We provide new results for RISE on MBPP & CoNaLa, two coding tasks in Table 3 in the 1-page PDF. We were able to only run 1 iteration on top of a Llama2-7B model in the rebuttal period, and find that RISE obtains better 5-turn performance w/o oracle (m1@t5) compared to 1-turn performance (m1@t1) by over 7.0% on MBPP and 1.7% on CoNaLa. RISE also attains higher sequential (m1@t5) performance vs parallel (m5@t1). We will run more iterations on top of this for the final version.
> ___
>
> ##  **[New result]** Reasons why having an improving trajectory can lead to better final outputs
>
> We now add more results to understand why training the model via RISE allows us to express more complex distributions better (Reason 1 in Section 5). Concretely, we track the average training negative log-likelihood loss (NLL) values for the oracle response $y^*$ given the input prompt $x$ marginalized over intermediate steps in a multi-turn rollout, and compare it against the NLL values $-\log p_\theta(y^*|x)$ attained by directly attempting to predict the final response in Fig 1 of the 1-page PDF (labeled as ``Classic''). We find that for any given number of epochs (including fractional number of epochs on the x-axis), the NLL value is lower when conditioning on multi-turn data that RISE generates in comparison with oracle responses to the prompts obtained from an expert. This suggests that RISE is able to utilize the computation of tokens from previous turns to better fit the target distribution. We also measured the average NLL loss on all samples through training, sampled i.i.d. from the training dataset for RISE and classic fine-tuning and observed a similar trend in Figure 8 of the submission.
>
> To verify Reason 2 in Section 5, that training via RISE is able to implicitly induce a notion of correctness into the model, we already showed in the submission (Figure 7) that training with RISE on one dataset allows it to improve self-improvement performance on other out-of-distribution prompt distributions. **In the 1-page PDF, we now present a more complete and quantitative version of this result (Table 2 in the 1-page PDF).** If RISE were not learning an implicit model of what makes a response correct, we would expect it to not work well when faced with out-of-distribution prompts where error patterns would be significantly different than the training data. We are happy to perform more probing experiments to further quantify these, if the reviewer has any suggestions.
> ___
>
> ##  Is it possible to directly perform RL over the trajectories? Note that this is essentially an in-context RL process.
>
> This is a great question! In fact, we already discuss running online RL as an avenue for future work in Sec 7. We do not see this as impossible, but the reason why we were not able to do it was due to the absence of a stable multi-turn on-policy RL codebase that could run fast enough on our computational resources. Most existing LLM RL implementations focus on single-turn RLHF training, like TRL, HALOs, and LlamaFactory, and while there are some (e.g., ArCHer[3], ILQL[4]) that focus on multi-turn RL, we could not find a scalable setup for training 7B models, as these prior works largely train much smaller models. If you have suggestions for good codebases, we would absolutely try it for the final!
> ___
>
> ## Related works
>
> Thanks for pointing out these related works! Indeed, in-context RL and RISE are very related. We will add a discussion of these works and cite them in the related work section. To briefly discuss the relation, in-context RL also aims to produce an improved action in a trajectory when conditioned on past trajectories. However, there are several differences: **(1)** most literature in in-context RL that we are aware of focuses on non-LLM problems, while we focus on LLM problems, and for the first-time show that self-improvement of this form is possible in LLMs, **(2)** Unlike [2] that only attempts to predict the optimal action, we find that training on a mixture of optimal and suboptimal data is more useful in improving performance., **(3)** While these prior works mainly focus on results showing that their approach performs well, we also present results in Section 5 to understand why we can get self-improvement at all in the first place.  **(4)** Finally, we also note that in-context RL utilizes the structure of an RL trajectory, since they predict the action given the current state, prior steps in the same trajectory, and past trajectories, but we do not utilize any such structure within an LLM response, we directly predict a _full_ response conditioned on the past responses, with no external feedback beyond a generic prompt that asks the model to retry. This makes our setting significantly more challenging than in-context RL where external feedback in the form of environment state is available after each action, _within_ the same rollout. We will add this discussion in the paper.
> ___
>
> References
>
> [3] Zhou, Yifei, et al. "Archer: Training language model agents via hierarchical multi-turn rl." arXiv:2402.19446 (2024).
>
> [4] Snell, Charlie, et al. "Offline rl for natural language generation with implicit language q learning." arXiv:2206.11871 (2022).

---

> > ### Comment · Reviewer_BtCg · 2024-08-12
> > **Thanks**
> >
> > The new results look great to me!

---

> > > ### Author Response · Authors · 2024-08-12
> > > **Thank You!**
> > >
> > > Thank you so much for the reply! We are glad that the new experiments look great!!

---

### Official Review · Reviewer_spvE · 2024-07-12

**Soundness:** 3
**Presentation:** 3
**Contribution:** 3
**Rating:** 7
**Confidence:** 3

**Summary:**

This paper introduces RISE: Recursive IntroSpEction, a novel approach to fine-tuning Large Language Models (LLMs) for self-improvement. The core idea is to enable LLMs to introspect, reason, and correct their mistakes over multiple turns. This is achieved by treating the fine-tuning process as a multi-turn Markov decision process (MDP), where the initial state is the prompt. The RISE algorithm iteratively fine-tunes the model using a combination of on-policy rollouts and reward-weighted regression (RWR) objectives. The method is shown to significantly improve the performance of 7B Llama2 and Mistral models on math reasoning tasks, outperforming several single-turn strategies.

**Strengths:**

- This paper presents a novel method for enabling self-improvement in LLMs, addressing a significant limitation in current models.
- The use of iterative fine-tuning and on-policy rollouts is a robust approach to teaching models how to correct their own mistakes.
- The approach is designed to be general, enabling self-improvement across a diverse set of problems and scenarios.

**Weaknesses:**

- While the results on GSM8K and MATH are promising, additional experiments on other types of tasks (e.g., natural language understanding, code generation) could strengthen the paper.
- The success of RISE may depend heavily on the initial quality of the model. Models that are already strong may benefit more from this approach than weaker models.

**Questions:**

1. In Section 4.2, it is mentioned that “(3) it must not contain any rollout that degrades in a subsequent turn. Our data collection strategy that satisfies these desiderata.” How is performance degradation determined? Does it require manual judgment?
2. Does “starting from a boosted model” introduce unfairness? Because the article does not mention whether other baselines undergo SFT (or maybe I missed it).
3. In Chapter 7, it is mentioned that “RISE requires running manual iterations.” What does "manual" mainly refer to here?
4. How should Figure 2 be interpreted? Why is the Success rate for Iteration2 Model lower than the Boost Model?
5. In Appendix C.2, it is mentioned “To control the randomness,” however, setting temperature to 1.0 and top_p to 1.0 seems counterintuitive?
6. How does the model locate errors in Figure 11? “the model is able to locate the error and perform local computation correctly.”
7. In Figure 4, what does on-policy+Expert (best of n) mean? The article mentions “Best-of-N” means using the best sample out of N from the learner (here N = 16), which seems to conflict with expert? Or can the learner here refer to the expert? And what does Expert+Expert mean? Is it related to the “DAgger [34]”-style approach?
8. If the reward function is a sparse binary indicator (0 or 1), how does it reflect the superiority of Reward-weighted RL? What is the difference between it and “simply imitating filtered successful data”?
9. Typos: figure 1 right, turn 1 response -> turn 2 response.

**Limitations:**

The author has already discussed the limitations ofthe method and its potential impacts in the paper, and provided possible solutions.

---

> ### Author Rebuttal · Authors · 2024-08-07
>
> Thank you for your feedback and for a positive assessment! To address your concerns, we **add new results below on MBPP[1] and CoNaLa[2] (Table 3 in 1-page PDF), two coding benchmarks that show the efficacy of RISE on coding tasks**. We also present results showing the efficacy of RISE with weak models (Table 6), weak-to-strong generalization (Table 5), and comparison against imitating optimal/suboptimal data  (Table 4). We answer your questions and will update the paper with these. **Please let us know if our responses have addressed your concerns, and if so, we would be grateful if you might be willing upgrade your score.**
>
> ___
>
> ## **[New results]** W1: Results on other tasks (e.g. code generation)
>
> We provide new results for RISE on MBPP & CoNaLa, two coding tasks in Table 3 in the 1-page PDF. We were able to only run 1 iteration on top of a Llama2-7B model in the rebuttal period, and find that RISE obtains better 5-turn performance w/o oracle (m1@t5) compared to 1-turn performance (m1@t1) by over 7.0% on MBPP and 1.7% on CoNaLa. RISE also attains higher sequential (m1@t5) performance vs parallel (m5@t1). We will run more iterations on the final.
> ___
>
> ## **[New results]** W2: The success of RISE may depend heavily on the initial quality of the model.
>
> Of course, if the model has no ability to propose meaningful responses, then it will not benefit from RISE. That said, we found that an initial round of boosting (SFT on in-domain data) was sufficient for making a model amenable to RISE. After boosting, RISE improves the performance of weaker models by a larger %age vs stronger models (see Table 6 in 1-page PDF).
>
> Since the submission, we also studied the weak-to-strong generalization capabilities of RISE (Burns et al. 2023), and found that multi-turn rollouts from weak models (e.g., Llama-2) can be very useful for RISE training of strong models (e.g., Mistral), indicating that even a model of worse quality can be used to generate data for training (Table 5).
> ___
>
> ## **[New result]** 9. Comparison against imitating filtered data
>
> We added a result in Table 4 in the PDF comparing RISE w/ running single-turn SFT on optimal + suboptimal data, and find RISE to be better. We already show that multi-turn training with only filtered data does poorly in Fig 3 of the submission.
>
> ___
>
> ## Questions
>
> > 1. In Section 4.2, how is performance degradation determined? Does it require manual judgment?
>
> By “performance degradation”, we refer to the case when an action in a subsequent turn attains lower reward than in the previous one. This does not involve manual judgment, since we  do an automated string match of the answer with the oracle to evaluate reward.
>
> > 2. Does “starting from a boosted model” introduce unfairness?
>
> For all comparisons we run (RISE, self-refine w/ 7B), we used knowledge boosting. Other comparisons (Eurus, GLoRE, >7B) borrowed from prior work also have similar phases, for e.g.,  Eurus runs SFT on an even larger math dataset than what we use; GLoRE runs SFT on the task data before training for refinement (Section 4 in GLoRE).
>
> > 3. “RISE requires running manual iterations.” What is "manual"?
>
> “Manual” here means we need to collect data and rerun training separately as two jobs. If we switch to a fully online RL variant of RISE or structure our code in a way that one script could launch data collection followed by training, then it will be fully automated.
>
> > 4. Figure 2:  Why is the Success rate for Iteration2 lower than the Boost?
>
> We apologize for the short caption, which we believe is the reason for the confusion. **Both bars in this figure plot the success rate of the Iteration 2 model**, but the rates are computed over the set of problems that the Boost model (green) or the Iteration 2 model (orange) get wrong within B parallel samples (x-axis) in turn 1. Orange bar will be lower as the problems not solved by Iteration 2 model are harder.
>
> > 5. In App. B.2, “To control the randomness,” however, setting temperature to 1.0 and top_p to 1.0 seems counterintuitive?
>
> We will change the phrase "control the randomness" to “modulate the stochasticity”. We did not mean that there is no randomness in sampling but rather that we could obtain stochasticity by changing temperature to 1.0.
>
> > 6. How does the model locate errors in Fig. 11?
>
> We apologize for the loose wording. In Fig. 11, each subsequent turn makes targeted step-level edits to the response. In particular, from turn 1 to turn 2, it changes Step 4 to make it correct (Step 4 / turn 2, in green). It does so while not changing the other steps in the response. It does  identify the wrong step in turn 1 correctly. We therefore called it “error location and editing”. That said, we agree this phrasing is a bit confusing, and will remove this phrasing.
>
> > 7. Fig 4 legend; relation to DAgger.
>
> The legend should be interpreted as <method for generating turn 1 response> + <method for generating turn 2 response>. For e.g., "on-policy + Expert (best-of-n)" means that we sample one from the learner, then sample $N$ from an expert and pick the best. Both first and second turn responses are generated from the expert in "Expert + Expert".  This is related to DAgger, as it shows that on-policy samples in turn 1 followed by expert does better than Expert + Expert.
>
> > 8. If the reward function is a sparse...difference between it and “simply imitating filtered successful data”?
>
> The reward is used to provide a weight $\exp(r/\tau)$ with $\tau = 0.4$. Thus RISE trains on both optimal (w/ larger weights) and suboptimal data, instead of optimal data only.
>
> > 10. Typos: Fig 1 R, turn 1 response -> turn 2 response.
>
> Thanks for pointing this out, we will fix this typo.
>
> ___
>
> References
>
> [1] Austin, Jacob, et al. "Program synthesis with large language models." arXiv preprint arXiv:2108.07732 (2021).
>
> [2] Yin, Pengcheng, et al. "Learning to Mine Aligned Code and Natural Language Pairs from Stack Overflow." arXiv preprint arXiv:1805.08949 (2018).

---

> ### Comment · Reviewer_spvE · 2024-08-12
>
> Many thanks to the authors for thoroughly supplementing the key experiments and discussions. I have raised the score to 7.

---

> > ### Author Response · Authors · 2024-08-12
> > **Thank you for raising your score!**
> >
> > Thank you so much for your reply! We are glad that your concerns are addressed and are grateful to you for the score increase.

---

### Official Review · Reviewer_yTdP · 2024-07-15

**Soundness:** 3
**Presentation:** 3
**Contribution:** 3
**Rating:** 5
**Confidence:** 4

**Summary:**

The manuscript tries to solve the problem that existing large language models (LLMs) dono't have the ability of continually improving their responses sequentially, even in scenarios where they are explicitly told that they are making a mistake. The authors propose a fine-tune approach, so-called RISE (Recursive IntroSpEction), that aims to enhance the self-improvement capabilities of LLMs to tackle these complex problems. The RISE method refines LLMs through an iterative fine-tuning process that teaches them to improve their responses following additional environmental feedback. It reformulates the single-turn fine-tuning problem as a multi-turn Markov decision process (MDP), and trains using a reward-weighted regression (RWR) objective. During inference, RISE enables models to detect and correct their previous mistakes over multiple iterations, thereby incrementally enhancing their problem-solving abilities. The results show that LLMs trained via RISE can produce correct responses on more prompts, improving over turns for more challenging prompts.

**Strengths:**

The RISE algorithm, which leverages principles from online imitation learning to improve the LLM's capacity for self-improvement over multiple turns of interaction, and queries expert supervision on states attained by on-policy rollouts.
It poses fine-tuning for a single-turn problem as solving a multi-turn Markov decision process (MDP), where the initial state is the prompt.
The learner is fine-tuned using a reward-weighted regression objective, which is able to learn from both high- and low-quality parts of rollouts.
Experimental results demonstrate that, RISE can enable similarly-sized LLMs to with self-improvement capabilities, with a notable advancement in monotonically increasing task performance in mathematical reasoning tasks.

**Weaknesses:**

1. In the MDP formulation, the action is denoted as the output response of the foundation models. This means that the action space would be very large, which make the MDP hard to learn.
2. Three conditions in the data collection, need a lot of sequentially prompts or human feedbacks. In the experiments, the scale or number of iterations is somehow small, corresponding to the self-improvement problem.
3. The equation 4.6 indicates that the RISE is actually a sequentially weighted supervised learning, and the RL/MDP model is confused with the original LLMs.

**Questions:**

1. How many training data is collected and used?
2. The reward is 1 if and only if a = y*. How to judge the reward when collecting data, when the LLM give an improving answer but still not right one?
3. In equation 4.4-4.6, the LLM is treated as a offline RL Model with weighted reward genereted. It might be better to provide more theoretical details.

**Limitations:**

Plenty of experimental results are presented. However, there are some computational constraints, that all the iterations is less than 3, and the turn is fixed as 5.  The scale or number of iterations in the experiment is somehow small, corresponding to the self-improvement problem.

---

> ### Author Rebuttal · Authors · 2024-08-07
>
> Thank you for your feedback and for a positive assessment! To address your concerns, we have added a new result running RISE for > 5 turns (Table 1 in the 1-page PDF), and find that RISE still continues to outperform other methods. We address your questions below & will update the paper. **Please let us know if your concerns are addressed, and if so, we could be grateful if you could raise your score.**
> ___
>
> ## **[New result]** Limited number of turns.
>
> To address this concern, we present a new result on GSM8k with Llama-2-7B (Table 1 in the 1-page PDF) that runs RISE for 10 turns. Following the paper, we compute: **mk@t1**: majority over $k$ responses sampled at the first turn w/o oracle; **m1@tk**: majority voting over $k$ sequential turns w/o oracle; and **p1@tk**: k-turn performance w/ oracle. Note that with more turns (rows), **RISE consistently improves with more turns (m1@tk increases as k increases) and exhibits m1@tk > mk@t1**. We are computing this result for MATH, and will add it to the final.
>
> That said, note that 5-turns in the submission is already larger than prior work that studies LLM self-improvement, for example, preference trees[1] (7B SOTA on math) considers 5 turns; self-refine[2] considers 3 turns, and GLoRE[3] considers 2 turns.
>
> ___
> ## Limited number of iterations
>
> Indeed, compute is a constraint for us. Collecting data and running one iteration of finetuning resulted in a cycle of 24 hours/iteration on our hardware, which led us to do 3 iterations (boost + 2 iterations) but add breadth to our results with more ablations. In fact, this is consistent with prior work that runs upto 3 iterations on GSM8K / MATH: Rest-EM[4], V-STaR[5] run only 3 iterations; and GLoRE[3], Eurus[2] only run 1 iteration. We will run at least 5 iterations for the final.
>
> ___
>
> ## Questions
>
> > In the MDP,….. the action space would be very large, which makes the MDP hard to learn.
>
> There is no difference between the notion of an action in standard LLM SFT/RLHF and the notion of an action in our MDP formulation, i.e., the response for a prompt. Our MDP gives the ability to improve actions by taking new actions. This should only intuitively make it easier to learn to produce correct answers than SFT. We confirm this in our new result in Figure 1 (1-page PDF) and Sec. 5 where RISE attains smaller training negative log likelihood (NLL) loss of the optimal y* given a prompt x.
>
> > Three conditions in the data collection, need a lot of sequentially prompts or human feedbacks.
>
> We clarify that there is no human feedback utilized in any experiment: all sequential on-policy data collection is automated. A fixed prompt (Appendix B.4) is used to ask the model to retry, so no prompt engineering is used either. While we do agree that RISE requires running on-policy rollouts, this is very similar to any on-policy / online RL approach for fine-tuning LLMs.
>
> > The equation 4.6 indicates that the RISE is actually a sequentially weighted supervised learning, and the RL/MDP model is confused with the original LLMs.
>
> We apologize if we are misunderstanding this question. Note that sequential weighted supervised learning via reward-weighted regression is a well-accepted offline RL approach (see: AWR[6]), and it has been benchmarked in offline RL benchmarks (e.g., D4RL[7]). That said, we are happy to change the terminology to weighted supervised learning, if the reviewer thinks that’d be more appropriate here.
>
> > How much training data is collected and used?
>
> We present the details of the number of training data points in Appendix B.3. To clarify, GSM8K consists of 7473 problems and MATH consists of 7500 problems that we utilize for training. We generate 1 multi-turn rollout per prompt and at any iteration $i$, the length of this multi-turn rollout for any problem is at most $i+1$.
>
> > The reward is 1 if and only if a = y*. How to judge the reward when collecting data, when the LLM give an improving answer but still not right one?
>
> We simply use a string match between the final answer and the expected answer to determine the binary 0-1 reward. Note however, that even when the reward is 0, our approach still trains on this data as Eq 4.6 utilizes $\exp(r / \tau)$ as the weight multiplier.
>
>
> > In equation 4.4-4.6, the LLM is treated as a offline RL Model with weighted reward generated. It might be better to provide more theoretical details.
>
> We would like to seek for a clarification as to what sorts of theoretical details would be appropriate to add in the paper. Prior work has shown that reward-weighted regression (or more concretely, advantage-weighted regression) is a consistent surrogate objective for maximizing return in an MDP, which we set as our main objective in Sec. 3 (see Page 3-5 of [6]). This derivation will apply as is to our setting as well. We are happy to add this derivation or add other theoretical details in the paper, if the reviewer has suggestions. Please let us know what would be most appropriate.
>
> ___
>
> References
>
> [1] Yuan, Lifan, et al. "Advancing llm reasoning generalists with preference trees." arXiv preprint arXiv:2404.02078 (2024).
>
> [2] Madaan, Aman, et al. "Self-refine: Iterative refinement with self-feedback." Advances in Neural Information Processing Systems 36 (2024).
>
> [3] Havrilla, Alex, et al. "Glore: When, where, and how to improve llm reasoning via global and local refinements." arXiv preprint arXiv:2402.10963 (2024).
>
> [4] Singh, Avi, et al. "Beyond human data: Scaling self-training for problem-solving with language models." arXiv preprint arXiv:2312.06585 (2023).
>
> [5] Hosseini, Arian, et al. "V-star: Training verifiers for self-taught reasoners." arXiv preprint arXiv:2402.06457 (2024).
>
> [6] Peng, Xue Bin, et al. "Advantage-weighted regression: Simple and scalable off-policy reinforcement learning." arXiv preprint arXiv:1910.00177 (2019).
>
> [7] Fu, Justin, et al. "D4rl: Datasets for deep data-driven reinforcement learning." arXiv preprint arXiv:2004.07219 (2020).

---

> > ### Author Response · Authors · 2024-08-13
> >
> > Dear Reviewer,
> >
> > As there are only a few hours remaining in the discussion period, we were wondering if you have gotten a chance to look at our response and if our responses address your questions. Especially, if you have any questions that might help improve your evaluation of our paper, we would love to answer them in the remaining time possible. Thanks so much!

---

> > ### Comment · Reviewer_yTdP · 2024-08-14
> >
> > Thanks to the authors for thoroughly supplementing the key experiments and explanation. I am glad to raise the score to 6.

---

> > > ### Author Response · Authors · 2024-08-14
> > >
> > > Thank you for your positive feedback and for raising the score to 6. We greatly appreciate your thorough review and the time you've taken to consider our supplementary materials and explanations.
> > >
> > > We noticed that while you mentioned raising the score to 6 in your comments, this update is not yet reflected in the official review. Would you kindly update the score in the official review as well? This will ensure that your revised assessment is accurately recorded for the paper evaluation process.
> > >
> > > Thank you once again for your valuable input and for your attention to this matter.

---

### Author Rebuttal · Authors · 2024-08-07

We thank the reviewers for their detailed feedback and reviews! We are glad that all the reviewers had a positive assessment of our work and we believe that addressing the reviewers’ feedback in this rebuttal period has made the paper stronger.

**We have added several new empirical results** in the rebuttal period (please see attached 1-page PDF), which include experiments highlighting the efficacy of RISE on **two** coding benchmarks (MBPP and CoNaLa), comparisons of RISE and parallel sampling for 10 turns (double the number of turns in the original paper); comparisons against simply imitating filtered data; weak-to-strong generalization; and experiments towards understanding why self-improvement with RISE is possible. We also made several clarifications pertaining to the MDP structure, scale of our experiments, clarifications regarding figures, baselines, and reward functions, and we appreciate the suggestions for related work. We will update the final version of the paper to include these clarifications and explanations.

In this global response, we summarize these experiments along with the reviewer they are intended towards, and the main results that we show in the 1-page PDF. **We look forward to the discussion!**

- **Table 1** (Reviewer yTdP): The performance of RISE with Llama-2-7B over 10 turns on GSM8K

- **Table 2** (Reviewer BtCg) Out-of-distribution prompt generalization for RISE

- **Table 3** (Reviewer spvE, BtCg)The performance of RISE on MBPP and CoNaLa.

- **Table 4** (Reviewer spvE): RISE vs. single-turn SFT (successful / unsuccessful data)

- **Tabel 5** (Reviewer spvE): Weak-to-strong generalization for RISE on GSM8K

- **Table 6** (Reviewer spvE): The percentage of improvement by RISE on GSM8K

- **Figure1** (Reviewer yTdP, BtCg): Negative log-likelihood (NLL) of the optimal response $y^∗$ given a prompt $x$ over training.

---

> ### Author Response · Authors · 2024-08-12
>
> Dear Reviewers,
>
> Apologies for bothering you! Since we are getting close to the end of the discussion period (less than two days remaining), we would be grateful and would sincerely appreciate if you could respond to our rebuttal including several new results on more turns, code generation domains, and understanding experiments to validate why self-improvement is possible (please see above), leaving us enough time to address any remaining questions.
>
> Best,
> Authors

---

### Comment · Area_Chair_NNT5 · 2024-08-12
**Discussion Period**

The discussion period is almost over, so both authors and reviewers please respond to any unaddressed questions. Reviewers, be sure that you have all of the information you need from the authors, since after the 13th, they won't be able to respond.

---

### Decision · Program_Chairs · 2024-09-25

**Decision:**

Accept (poster)

**Comment:**

While the reviewers initially had some concerns around implementation details and the focus on math and reasoning benchmarks, the authors managed to clarify things significantly and even added an additional result which broaden the overall scope of the work.